# Menin in Cancer

**DOI:** 10.3390/genes15091231

**Published:** 2024-09-21

**Authors:** Ariana D. Majer, Xianxin Hua, Bryson W. Katona

**Affiliations:** 1Division of Gastroenterology and Hepatology, Perelman School of Medicine, University of Pennsylvania, Philadelphia, PA 19104, USA; majerar@pennmedicine.upenn.edu; 2Department of Cancer Biology, Perelman School of Medicine, University of Pennsylvania, Philadelphia, PA 19104, USA; huax@pennmedicine.upenn.edu

**Keywords:** menin, cancer, nuclear scaffold protein, tumor suppressor, tumor promoter

## Abstract

The protein menin is encoded by the *MEN1* gene and primarily serves as a nuclear scaffold protein, regulating gene expression through its interaction with and regulation of chromatin modifiers and transcription factors. While the scope of menin’s functions continues to expand, one area of growing investigation is the role of menin in cancer. Menin is increasingly recognized for its dual function as either a tumor suppressor or a tumor promoter in a highly tumor-dependent and context-specific manner. While menin serves as a suppressor of neuroendocrine tumor growth, as seen in the cancer risk syndrome multiple endocrine neoplasia type 1 (MEN1) syndrome caused by pathogenic germline variants in *MEN1*, recent data demonstrate that menin also suppresses cholangiocarcinoma, pancreatic ductal adenocarcinoma, gastric adenocarcinoma, lung adenocarcinoma, and melanoma. On the other hand, menin can also serve as a tumor promoter in leukemia, colorectal cancer, ovarian and endometrial cancers, Ewing sarcoma, and gliomas. Moreover, menin can either suppress or promote tumorigenesis in the breast and prostate depending on hormone receptor status and may also have mixed roles in hepatocellular carcinoma. Here, we review the rapidly expanding literature on the role and function of menin across a broad array of different cancer types, outlining tumor-specific differences in menin’s function and mechanism of action, as well as identifying its therapeutic potential and highlighting areas for future investigation.

## 1. Introduction

Menin, encoded by the *MEN1* gene located on chromosome 11q13, is a unique protein consisting of 610 amino acids and weighing 67 kDa [1,2] that lacks homology with proteins in lower organisms [3]. Menin is expressed in all tissues [4,5] and is essential for proper development, as the homozygous deletion of *Men1* in mice is embryonic lethal at E11.5–13.5 as a result of defects in multiple organs [6]. Menin lacks domains that are homologous to other proteins [1,2], and menin itself lacks any enzymatic activity [7]. Consequently, menin primarily functions as a nuclear scaffold protein, and its effects are mediated through interactions with many partners. Structurally, menin contains three nuclear localization signals in its C-terminal region and primarily localizes to the nucleus [8,9,10]; however, menin can also localize to the cytoplasm [11,12], primarily during mitosis [12,13], and even to the cell membrane [14,15,16]. The tertiary structure of menin forms a central binding pocket that facilitates interactions with over 50 proteins [17], including transcription factors, chromatin modifiers, cell cycle proteins, DNA repair proteins, and cell signaling proteins [17,18,19]. Menin interacts with different proteins in different tissues and contexts, and menin’s functions are, therefore, highly tissue- and context-specific [18]. In the context of cancer, menin can function as either a tumor suppressor or a tumor promoter, depending on the tissue type and context (Figure 1). Here, we review the role of menin across various cancers and outline the tumor-specific differences in menin’s function and mechanism of action, as well as highlight the therapeutic potential of menin inhibitors and areas for future investigation.

## 2. Neuroendocrine Tumors

The *MEN1* gene was first discovered in the context of multiple endocrine neoplasia type 1 (MEN1) syndrome [1,2], where it functions as a potent tumor suppressor gene (reviewed in [7,20,21,22,23]). Individuals with MEN1 syndrome are born with a pathogenic germline variant of the *MEN1* gene and are at increased risk of developing neuroendocrine tumors (NETs). These NETs primarily develop in the endocrine pancreas, parathyroid, and pituitary following loss of heterozygosity (LOH) of *MEN1*, but loss of menin function is also associated with increased risk of developing luminal gastrointestinal, lung, and thymic NETs (reviewed in [7,20,21,22,23,24]). Heterozygous ablation of menin in mice similarly results in tumor formation in neuroendocrine tissues reminiscent of human MEN1 syndrome (reviewed in [20,22,24,25]). Menin is also frequently mutated in sporadic NETs [24,26,27,28,29,30,31,32], highlighting that loss of menin expression is a key driver of neuroendocrine tumorigenesis. Notably, of all the tissues in which menin regulates tumorigenesis, neuroendocrine tissues are the only ones in which menin is consistently mutated [22]. These mutations result in mislocalization, altered protein–protein interactions, and instability [11,25,33,34,35,36,37], typically rendering the mutant protein nonfunctional. While *MEN1* LOH is considered necessary for neuroendocrine tumorigenesis, recent data suggest that epigenetic downregulation of menin expression, such as by miRNAs, may be sufficient to initiate tumorigenesis prior to LOH (reviewed in [38,39]). Herein, we will explore the tumor-specific roles of menin in five subtypes of NETs: pancreatic NETs, parathyroid tumors, pituitary tumors, lung NETs, and luminal gastrointestinal NETs.

### 2.1. Pancreatic NETs

#### 2.1.1. Cell Cycle Control and Genome Stability

Inactivating *MEN1* mutations are the predominant genetic defect driving tumorigenesis in pancreatic NETs (pNETs) [20]. Loss of menin promotes aberrant proliferation of pancreatic β-cells through the loss of cell cycle control [21,22,23,24,25,26]. It is well established that menin inhibits cell cycle progression by promoting the expression of the G1/S phase transition cell cycle inhibitors p27 (encoded by *CDKN1B*) and p18 (encoded by *CDKN2C*) in multiple neuroendocrine tissues [21,22,23,24,27,28,29,30,31]. While there are some conflicting data on menin’s regulation of p27 in the endocrine pancreas specifically [32,33], multiple studies demonstrate that menin promotes p18 and p27 expression in islet β-cells and islet tumors, as p18 and p27 are downregulated in menin-deficient islets of both *Men1*^+/−^ and *Men1*^f/f^; Cre-ER mice compared to menin wild-type islets [21,24]. Mechanistically, menin binds to the *CDKN1B* and *CDKN2C* promoters and increases histone 3 lysine 4 (H3K4) trimethylation [24], presumably through its interaction with MLL (also known as KMT2A), as knocking out both Mll and menin more strongly promotes β-cell proliferation and accelerates islet tumor formation [34]. Menin can also inhibit cell cycle progression by inhibiting the expression of the G1 cyclin-dependent kinase CDK4 [21,24] and by repressing the activity of the activator of S phase kinase (ASK) [25]. CDK4 expression is upregulated in islet tumors of *Men1*^+/−^ mice and in hyperplastic islets of menin knockout mice (*Men1*^f/f^;Cre-ER) compared to islets from menin wild-type mice [21,24]. Loss of menin in pancreatic islets of mice markedly increases S phase entry [21] and cell proliferation as a result of increased ASK activity [25]. Furthermore, menin may also delay the G2/M phase transition in the endocrine pancreas, as well as in other neuroendocrine tissues [26]. Genetic depletion of menin in mouse embryonic fibroblasts (MEFs) increases proliferation by upregulating the expression of cyclin B2 (encoded by *Ccnb2*). Menin facilitates the binding of transcriptional activators, such as nuclear factor Y (NF-Y), E2 factors (E2Fs), and CREB-binding protein (CBP), at the *Ccnb2* promoter, subsequently increasing promoter acetylation [26]. Together, these data implicate menin as a key regulator of cell cycle control in the endocrine pancreas and indicate that menin inhibits pNET tumorigenesis by maintaining cell cycle control.

In addition to inhibiting cell cycle progression, menin may also suppress pNET tumorigenesis through the regulation of DNA damage and genome stability. Loss of menin function in pNETs and cell lines is associated with dysregulation of genes involved in DNA replication, DNA recombination, DNA repair, and telomere maintenance [35,36]. *MEN1* LOH in pNETs is also associated with loss of DNA integrity [37], chromosomal instability [37], and cytogenetic abnormalities [38], potentially due to defects in chromosome segregation. A recent study in HeLa cells found that menin localizes to the mitotic spindle during early mitosis and to the intercellular bridge microtubules during cytokinesis [13] and is necessary for proper spindle assembly and chromosome segregation [13]. Menin may also promote DNA repair through its interactions with the DNA repair proteins RPA2 [39] and FANCD2 [10] in response to DNA damage, though the precise nature of these interactions and the mechanisms by which they affect DNA repair in the endocrine pancreas remain unexplored.

#### 2.1.2. DNA Methylation

Menin may also suppress pNET development and progression by inhibiting DNA methylation in the endocrine pancreas. DNA is globally hypermethylated in MEN1-associated pNETs [40,41], and DNA methyltransferase 1 (DNMT1) expression is significantly upregulated in the pancreatic islets of MEN1 individuals and menin-deficient mice. Genetic depletion of menin in MEFs increases Dnmt1 mRNA levels and enzymatic activity [42]. Furthermore, overexpression of DNMT1 in *RIP-TVA* mice increases DNA methylation and subsequently increases β-cell proliferation, resulting in islet hyperplasia [43]. One gene that may be specifically methylated by menin in the endocrine pancreas is *CASP8* (encoding caspase-8). The *CASP8* promoter is hypermethylated in MEN1-associated pNETs compared to sporadic pNETs [41]. This hypermethylation leads to downregulation of caspase-8 expression [44], as re-expressing menin in *Men*^−/−^ MEFs restores caspase-8 expression [45]. Moreover, multiple *MEN1* mutations associated with MEN1 syndrome abolish menin’s ability to activate caspase-8 expression [46]. Notably, the downregulation of caspase-8 suppresses tumor necrosis factor α (TNF-α)-induced apoptosis [44,45], suggesting that menin can also suppress pNET tumorigenesis by sensitizing pancreatic islets to apoptosis. In MEN1-associated pNETs, the loss of menin also leads to hypermethylation of the tumor suppressor genes *RASSF1* and *APC*, as well as the *MGMT* gene, which encodes a DNA repair enzyme [41,47]. Furthermore, Tirosh and colleagues reported that DNA hypermethylation in MEN1-associated pNETs is enriched at genes involved in VEGF signaling, insulin secretion regulation, and the phosphatidylinositol-4,5-bisphosphate (PIP2) pathway [40]. These genes and pathways suggest that menin may suppress pNET tumorigenesis by silencing the transcription of key tumor suppressor genes; however, the functional significance of the hypermethylation of these genes in pNETs remains unexplored.

#### 2.1.3. Hedgehog and Wnt/β-Catenin Signaling

Menin also suppresses pNET tumorigenesis by inhibiting Hedgehog signaling and subsequent activation of downstream pro-proliferative genes [48]. Menin interacts with the histone arginine methyltransferase PRMT5 [49] and promotes the placement of repressive H4R3me2 marks at the promoter of growth arrest-specific 1 (*GAS1*) [49,50], which encodes a protein that is necessary for the activation of the Hedgehog signaling pathway [48]. Loss of menin in the pancreatic islets of mice results in the upregulation of Gas1 expression and activates the expression of Gli1, a downstream effector of Hedgehog signaling [48]. Furthermore, treating *Men1*^f/f^;*RIP*-Cre mice with the Hedgehog inhibitor GDC-0449 markedly reduces the proliferation of β-cells and reduces insulin secretion by the insulinomas that form [49]. Together, these data indicate that menin suppresses pNET tumorigenesis by inhibiting Hedgehog signaling through PRMT5-mediated repression of *GAS1*.

Menin may also suppress pNET tumorigenesis by inhibiting Wnt/β-catenin signaling [8,20,51,52]. Loss of menin increases the nuclear accumulation of β-catenin in menin-null MEFs and insulinomas that form following β-cell-specific menin knockout compared to menin wild-type MEFs and menin-expressing pancreatic islets, respectively [8]. Mechanistically, menin directly interacts with β-catenin and facilitates β-catenin nuclear export in a CRM1-dependent manner [8]. Inhibiting Wnt/β-catenin signaling in *Men1*-deficient murine β-cells reduces proliferation [20], and combining a β-cell-specific menin knockout with a β-catenin knockout decreases tumor formation compared to menin knockout alone [20]. These data indicate that menin inhibits Wnt/β-catenin signaling in islet β-cells and suggest that pharmacologic inhibition of Wnt/β-catenin may be a viable therapeutic strategy for *MEN1* LOH pNETs.

#### 2.1.4. PI3K/Akt/mTOR Signaling

Menin may also suppress pNET tumorigenesis through inhibition of phosphatidylinositol 3-kinase (PI3K)/Akt/mTOR signaling pathway. The mammalian target of rapamycin (mTOR) signaling is frequently upregulated in MEN1-associated pNETs [53], and overexpressing menin in pNET cell lines decreases mTOR activation [54]. Upregulation of mTOR signaling promotes pNET cell migration [55] and inhibits ferroptotic cell death in pancreatic islets through the upregulation of the negative regulator of ferroptosis SCD1 [54,56]. Mechanistically, menin inhibits mTOR signaling in pancreatic β-cells by interacting with and inhibiting the activation of Akt [15,55], subsequently suppressing Akt-dependent proliferation while promoting apoptosis in murine β-cells [15,55]. Menin also interacts with the PI3K/Akt inhibitor phosphatase and tensin homolog (PTEN) in β-cells [57,58], suggesting that menin may also inhibit the activation of mTOR signaling by activating PTEN. Together, these data suggest that menin inhibits PI3K/Akt/mTOR signaling at multiple levels and thereby inhibits pNET development and progression by suppressing β-cell migration and promoting β-cell death.

#### 2.1.5. Differentiation Factors, Transcription Factors, and Growth Factors

Menin may also inhibit pNET development by regulating the expression of β-cell differentiation factors HLXB9, MAFA, MAFB, FOXA2, and NKX2.2. Hlxb9, MafA, MafB, Foxa2, and Nkx2.2 are differentially expressed in murine pancreatic islets following the loss of menin [59,60,61,62,63]. Menin binds to the promoters of *Hlxb9* [59,60] and *MafA* [61] in murine β-cells, and menin’s regulation of Hlxb9 and MafA suppresses β-cell proliferation [59,60,61] and promotes apoptosis [60]. Menin also suppresses pNET development by promoting the expression of insulin-like growth factor 2 mRNA-binding protein 2 (IGF2BP2) [46,64]. Genetic depletion of menin in the β-cells of mice decreases activating H3K4me3 and increases repressive H3K27me3 at the *Igf2bp2* promoter [64]. The decrease in H3K4me3 is likely mediated by the H3K4me3 demethylase retinoblastoma binding protein 2 (RBP2), as inactivation of *Rbp2* in *Men1* knockout mice reverses the downregulation of IGFBP2 in *Men1*-deficient islets [64]. Importantly, loss of *Rbp2* and restoration of Igf2bp2 expression decreases islet tumorigenesis and increases mouse survival [65]. Furthermore, menin can also suppress tumorigenesis of the endocrine pancreas through its interactions with the transcription factors CHES1/FOXN3 [66] and JunD [67] in β-cells and islet tumor cells. *MEN1* mutations that abolish menin’s interaction with either CHES1/FOXN3 or JunD are associated with increased malignancy, metastasis, and death [66,67]. Together, these data suggest that menin inhibits pNET tumorigenesis by regulating the expression of various differentiation, growth, and transcription factors that regulate β-cell proliferation, differentiation, and survival.

#### 2.1.6. Pleiotrophin

Menin also suppresses pNET tumorigenesis through inhibition of pleiotrophin (PTN) [68]. In a cohort of individuals with resected pNETs, low menin expression was positively correlated with high PTN expression. In addition, individuals with menin-low tumors that expressed PTN were more likely to exhibit metastatic disease and have shorter disease-free survival compared to their menin-low counterparts that did not express PTN [68]. Mechanistically, PTN increased pNET cell migration and invasiveness [68], potentially through the activation of FAK, PI3K, and ERK signaling [69]. However, unlike lung adenocarcinoma [69,70] and melanoma [71], genetic depletion of menin had opposing effects on PTN expression in two different pNET cell lines expressing wild-type menin [68], indicating that further research is needed to better understand the mechanisms by which menin regulates PTN in the endocrine pancreas.

#### 2.1.7. Non-Coding RNAs

Menin can also suppress pNET tumorigenesis by activating the transcription of the tumor-suppressive lncRNA *MEG3*. Menin promotes the expression of *Meg3* in embryonic stem cells and pNETs [72,73]. *MEG3* associates with the polycomb repressive complex 2 (PRC2) [74] and functions as a tumor suppressor in multiple different cell types by activating p53 target gene transcription and p53-induced apoptosis [75]. Overexpressing *Meg3* in a murine pNET cell line inhibits proliferation, delays cell cycle progression, and reduces cell migration and invasion through downregulation of the proto-oncogene c-MET [73]. MEN1-associated and murine menin knockout pNETs have lower expression of *MEG3* and higher expression of c-MET compared to non-malignant pancreatic islets [73], indicating that menin inhibits tumorigenesis in the endocrine pancreas through the positive regulation of *MEG3* expression. Menin may also suppress pNET tumorigenesis through the regulation of the miRNAs let-7a and miR-155. Menin interacts with the arsenite-resistance protein 2 (ARS2), which is a key component of the complex that stabilizes pri-miRNA. ARS2 facilitates the processing of pri-let-7a and pri-miR-155 and, ultimately, the biogenesis of mature let-7a and miR-155 [76]. Let-7a is generally recognized as a tumor suppressor [77], and miR-155 is frequently downregulated in pNETs [78], suggesting that menin can suppress pNET tumorigenesis by promoting the expression of these tumor-suppressive miRNAs.

Notably, menin is also regulated by miRNAs in the endocrine pancreas (reviewed in [79]). miR-17 targets the 3′-UTR of menin mRNA in response to hyperglycemia [80], downregulating menin protein levels and promoting β-cells proliferation [80]. miR-24-1 and miR-24-2 also recognize a sequence in the 3′-UTR of *MEN1* transcripts, promoting their degradation [81,82]. Importantly, the downregulation of menin following the expression of pre-miR-24 in immortalized pancreatic β-cells increases cell proliferation [81,82] as a result of decreased expression of p27 and p18 [82]. In addition to being regulated by *miR-24*, menin can also bind to the miR-24 pri-miRNA (pri-miR-24-1) in the endocrine pancreas and is required for both pri-miR-24-1 synthesis and the biogenesis of mature miR-24-1 [83,84], suggesting that menin can negatively regulate its own expression in the endocrine pancreas.

### 2.2. Parathyroid Tumors

Menin suppresses parathyroid tumorigenesis by inhibiting DNA methylation in the parathyroid, similar to its role in the endocrine pancreas. DNA is globally hypermethylated in MEN1 syndrome-associated parathyroid tumors [42] due to the upregulation of the DNA methyltransferase DNMT1 upon the loss of menin [42]. The genes frequently hypermethylated in menin-deficient parathyroid tumor tissue are most highly associated with the eukaryotic initiation factor 2 (EIF2), Wnt/β-catenin, oxidative phosphorylation, and granzyme A pathways [42]. With regard to Wnt/β-catenin signaling, multiple members of the Sox gene family are hypermethylated and downregulated in murine menin knockout tumors and in MEN1-deficient parathyroid tumors compared to benign parathyroid tissue [42]. Sox genes have previously been reported to inhibit the accumulation of β-catenin in the nucleus [42,85]. Accordingly, MEN1-associated parathyroid tissues exhibit increased expression of β-catenin [42], suggesting that the loss of menin can promote parathyroid tumorigenesis by indirectly activating Wnt/β-catenin signaling through hypermethylation and subsequent downregulation of Sox genes. Additional research is needed to characterize the functional significance of hypermethylation in genes related to eukaryotic initiation factor 2 (EIF2), oxidative phosphorylation, and granzyme A pathways in the tumorigenesis of menin-deficient parathyroid tumors.

Menin may also suppress parathyroid tumorigenesis through its regulation of Hox genes. Hox genes play a key role in the development of endocrine organs [86]. A study by Shen and colleagues identified 23 Hox genes whose expression was dysregulated in MEN1-associated parathyroid tumors compared to sporadic parathyroid tumors and non-malignant parathyroid tissue [87]. Notably, the majority of the dysregulated Hox genes were upregulated in MEN1-associated parathyroid tumors relative to non-malignant parathyroid tissue [87]; however, the functional significance of these changes in Hox gene expression in the parathyroid remains unexplored.

Menin also inhibits parathyroid cell proliferation through the regulation of transforming growth factor β (TGFβ) signaling. Exogenous TGFβ inhibits proliferation and parathyroid hormone (PTH) secretion in parathyroid cells derived from individuals with secondary hyperthyroidism. Importantly, the TGFβ-mediated decrease in proliferation and PTH secretion can be blocked by genetically depleting menin [88], and parathyroid cells derived from an individual with MEN1 syndrome do not respond to exogenous TGFβ [88]. These data suggest that menin can suppress parathyroid tumorigenesis by promoting TGFβ’s negative regulation of proliferation and hormone secretion.

Menin may also suppress parathyroid tumorigenesis by promoting the transcriptional activity of the vitamin D receptor. Menin interacts with the vitamin D receptor (VDR) in parathyroid cells and subsequently augments VDR activity and the activity of its interacting partner, the retinoid X receptor (RXR) [89]. VDR inhibits hormone secretion [90] and cell proliferation [91] in the parathyroid, potentially through the activation of its target genes such as *CYP24* [89]. CYP24 inhibits PTH secretion by repressing *PTH* transcription [92], and CYP24 is downregulated in *MEN1* mutant parathyroid adenomas [89,92]. However, the expression changes following menin inactivation have been assessed for only a handful of VDR target genes [89], and further research is necessary to better understand the role of menin and VDR in parathyroid tumorigenesis.

In addition, menin may suppress parathyroid tumorigenesis through the regulation of miRNAs, similar to its role in the endocrine pancreas. miR-4258, miR-664, and miR-1301, which are predicted to target mRNAs encoding the cell cycle proteins cyclin D1, p27, and p18, respectively, are differentially expressed in parathyroid adenomas with menin loss-of-function mutations [84]. Specifically, miR-4258 is markedly downregulated in parathyroid adenomas with *MEN1* LOH, while miR-1301 and miR-664 are upregulated in parathyroid adenomas with *MEN1* LOH or one inactivated *MEN1* allele, respectively [84]. These data suggest that loss of menin function in the parathyroid may promote aberrant cell growth through the loss of cell cycle control. In addition to regulating the cell cycle inhibitor p18, miR-664 is also predicted to target and downregulate *CDC73* [84], which has previously been shown to function as a tumor suppressor in the parathyroid [93,94]. Further functional studies are necessary to better understand the role of these miRNAs in parathyroid tumorigenesis and to determine whether menin suppresses tumorigenesis in any other endocrine tissues through the regulation of these miRNAs.

In addition to regulating miRNA expression, menin itself can be regulated by miRNAs in parathyroid tumors, similar to its role in pNETs. Both parathyroid adenomas and healthy parathyroid tissue show an inverse correlation between menin expression and expression of miR-24-1 [81]. Interestingly, this inverse correlation is present only when at least one copy of *MEN1* remains unmutated [81], suggesting that downregulation of menin by miR-24 may mimic the effect of a second somatic inactivating mutation in *MEN1* and initiate tumorigenesis prior to LOH. These data also suggest that antisense RNAs (antagomirs) targeting miR-24 may be effective therapeutic strategies for MEN1 patients with NETs who have not yet experienced LOH. Preclinical data in menin wild-type pNET cell lines showed that a miR-24 antagomir successfully inhibits miR-24 expression and increases menin expression [83].

### 2.3. Pituitary Tumors

Similar to its role in parathyroid tumors, menin suppresses pituitary tumorigenesis by inhibiting tumor cell proliferation through the activation of TGFβ signaling [95]. Menin interacts with the TGFβ-regulated transcription factor Smad3 in rat pituitary tumor cells, promoting the activation of TGFβ target genes that inhibit cell growth [95]. Knocking down menin in these cells disrupts Smad3 binding to DNA and inhibits Smad3-mediated transcriptional activation of the antiproliferative TGFβ target genes, subsequently promoting pituitary tumor cell growth [95]. Menin also suppresses pituitary tumorigenesis by inhibiting prolactin gene transcription. Overexpressing menin in rat pituitary cells downregulates the transcription of the prolactin gene, resulting in decreased cell proliferation [96]. Menin can further suppress pituitary tumorigenesis through the upregulation of tumor-suppressive miRNAs, similar to its role in pNETs. miR-15a, miR-16-1, and let-7a are downregulated in the pituitary tumors of Men^+/−^ mice compared to menin wild-type mice [97], and both miR-15a and miR-16-1 have been found to be significantly downregulated in human sporadic pituitary adenomas compared to non-malignant pituitary tissue [98]. Furthermore, knocking down menin in murine pituitary cells significantly decreases the expression of miR-15a [97]. Both miR-15a and miR-16-1 expression inversely correlated with the expression of cyclin D1 [97], and let-7a has been shown to function as a tumor suppressor in multiple tissues [77]. These data suggest that menin inhibits cell cycle progression in the pituitary through the upregulation of miR-15a and miR-16-1, though the mechanisms by which menin promotes the expression of these miRNAs remain unexplored.

### 2.4. Lung NETs

*MEN1* is frequently mutated in lung NETs [99,100,101,102,103,104,105,106,107], and ablation of one allele of *Men1* in mice accelerates lung NET development [108]. Interestingly, loss of menin in two mouse models resulted in the formation of lung tumors with high expression of the neuroendocrine markers neural cell adhesion molecule 1 (NCAM1), neuron-specific enolase, synaptophysin, and chromogranin A [108]. Whole-body knockout of Mll after birth similarly increases the expression of neuroendocrine markers in the lung [108], suggesting that menin suppresses lung NET development by suppressing neuroendocrine differentiation in the lung through its interaction with MLL. In a cohort of individuals with non-small cell lung cancer and a cohort with small cell lung cancer, low tumor menin expression was correlated with high expression of NCAM1, high expression of neuron-specific enolase, and shorter median survival time compared to high tumor menin expression [108]. These data further support the notion that loss of menin promotes neuroendocrine differentiation of the lung and suggest that increased neuroendocrine differentiation results in more aggressive tumors. Furthermore, these data suggest that there may be an overlap between menin’s role in lung adenocarcinomas and lung NETs, and further research is needed to determine whether the mechanisms by which menin suppresses lung adenocarcinoma are also at play in lung NETs. However, it is important to note that a study by Simbolo and colleagues suggests that menin may possibly promote lung NET progression in some contexts [109]. Simbolo and colleagues found that individuals with atypical lung carcinoids and large cell neuroendocrine carcinomas of the lung who had inactivating mutations in p53 and Rb but intact menin expression had shorter cancer-specific survival than individuals with inactivating p53 and menin mutations or intermediate features [109]. As this is the only study to suggest that menin may promote lung NET tumorigenesis, further study is needed to better understand the contexts in which menin may promote the development and progression of lung NETs.

### 2.5. Luminal Gastrointestinal NETs

Similar to pNETs, menin suppresses the development of gastric NETs through the activation of *CDKN1B* gene expression. Sundaresan and colleagues found that p27 is lost in menin-deficient murine gastric NETs as a result of both increased Skp2-mediated nuclear degradation of p27 and increased phosphorylation of p27, followed by nuclear export [110]. Loss of p27 expression is associated with decreased survival for luminal gastrointestinal NETs, as well as pNETs [111], suggesting that menin protects against gastrointestinal NET formation by maintaining cell cycle control.

Additionally, menin may suppress gastrinoma development by repressing gastrin expression. While there is no evidence to date indicating that loss of menin is sufficient to induce oncogenic transformation of gastric neuroendocrine cells [112,113,114,115], loss of menin in the gastric epithelium results in hypergastrinemia, as well as both neuroendocrine cell hyperplasia and epithelial dysplasia [115]. Mechanistically, menin inhibits gastrin expression by repressing JunD [116,117,118,119,120]. Menin binds JunD and prevents the activating phosphorylation of JunD by JNK [116,117,118]. Menin also cooperates with the general co-repressor mSin3A and its associated histone deacetylases (HDACs) to further inhibit the transcriptional activity of JunD [121,122], preventing JunD from activating gastrin expression [119,120]. Gastrin is a trophic peptide that promotes gastrointestinal cell proliferation (reviewed in [123]), suggesting that loss of menin may contribute to the initiation and progression of gastrinomas through increased gastrin expression.

Menin may also suppress luminal gastrointestinal NET formation partly through the repression of pro-proliferative genes. Menin interacts with the death-domain-associated protein 6 (Daxx) and activates the histone H3K9 methyltransferase SUV39H1 in a mouse intestinal NET cell line. Multiple MEN1-associated menin mutants abolish the interaction between menin and DAXX [52], suggesting that the inactivation of menin may promote tumorigenesis by altering DAXX activity. Mechanistically, activation of SUV39H1 by the menin/Daxx complex increases transcriptionally repressive H3K9 trimethylation of the promoters of genes associated with neuroendocrine tumorigenesis [52,124], including the promoter of membrane metalloendopeptidase (*Mme*) [52], which is associated with larger, more proliferative, and metastatic tumors [52,124].

Importantly, loss-of-function mutations identified in MEN1-associated gastroenteropancreatic NETs alter the subcellular localization of menin and destabilize the menin protein [11], rendering the mutant proteins unable to suppress cell proliferation and gastrin expression in gastroenteropancreatic cells [11]. Interestingly, the small molecule menin inhibitor MI-503 has been shown to stabilize some mutants and rescue nuclear localization in a mouse model of gastroenteropancreatic NETs [11]. In the mouse model, stabilization of the mutants by MI-503 rescued their tumor-suppressive activity, as measured by decreased hypergastrinemia and gastric hyperplasia in the mice [11]. These data indicate that some *MEN1* pathogenic germline variants promote tumorigenesis in neuroendocrine tissues by destabilizing nuclear menin expression and suggest that specific menin inhibitors may benefit MEN1 individuals by stabilizing and restoring the activity of the mutant proteins. It is unclear, however, whether the stabilization of these menin mutants is limited to MI-503, which is currently not approved for use in humans, or to specific *MEN1* mutations. Furthermore, it remains unclear which *MEN1* mutations are susceptible to stabilization and whether different menin mutants may be stabilized by different inhibitors. Future studies are, therefore, needed to better understand the potential utility of menin inhibitors as a therapeutic strategy to slow or even prevent the development and progression of NETs in individuals with MEN1 syndrome.

## 3. Leukemia

For decades, menin has been appreciated as a potent tumor promoter in MLL-rearranged (MLLr, also known as KMT2Ar) and nucleophosmin 1 (NPM1) mutant acute leukemias. Menin is necessary for maintaining leukemic transformation [125,126] through its interaction with both wild-type MLL and MLL fusion proteins and subsequent dysregulation of MLL target genes. One key family of genes regulated by the menin–MLL complex [127,128,129,130,131] and frequently dysregulated in leukemia (reviewed in [126,129,132]) is the Hox gene family, which plays a key role in hematopoiesis [127,128,130,131]. Hox gene expression is tightly controlled during normal hematopoiesis, with Hox gene expression decreasing during cell differentiation [133,134,135,136]. In MLLr and NPM1 mutant leukemias, however, Hox gene expression is aberrantly activated either by the MLL fusion proteins or by aberrant wild-type MLL activity mediated by NPM1 mutants, inhibiting differentiation and resulting in increased cell proliferation and survival that drives leukemogenesis [125,137,138,139,140,141,142,143,144,145]. Importantly, menin’s interaction with both wild-type MLL and MLL fusion proteins is necessary to maintain oncogenic activation of Hox genes [125,126,135,136,144,146,147,148], as well as the Hox cofactors MEIS1 [129,137,138,139,140,141,149,150,151,152,153] and PBX3 [141,151,153,154], and downstream targets such as FLT3 [129,137,141,150,153,155,156] in both MLLr and NPM1 mutant leukemias.

Along with its interaction with MLL, menin can also promote leukemia through its interactions with PSIP1, c-Myb, and DOT1L. Menin, along with the N-terminal portion of MLL, complexes with the transcriptional coactivator PSIP1 (also known as LEDGF/p75) [157]. The interaction between menin–MLL complex and PSIP1 is necessary for the leukemic transformation of MLL-ENL-expressing myeloid progenitor cells, primarily through the aberrant activation of the MLL target gene *HOXA9*, as well as *HOXA7* and *MEIS1* [157]. Menin also interacts with the transcription factor c-Myb [158]. Menin’s interaction with c-Myb is required for the recruitment of both wild-type MLL and MLL fusion proteins to target genes and is necessary for maintaining the hyperactivation of Hox gene transcription in both MLLr and MLL wild-type leukemias [158]. (For more details on the mechanistic interplay between menin and PSIP1 or c-Myb, please see [18,159,160]). In addition to PSIP1 and c-Myb, menin also interacts with the H3K79 histone methyltransferase DOT1L [161,162,163], forming a non-canonical oncogenic DOT1L complex [161]. Through this complex, menin and its associated oncogenic MLL fusion protein recruit DOT1L to MLL target genes (i.e., Hox genes), resulting in increased H3K79 methylation [162,163,164]. Notably, the increase in H3K79 methylation at MLL target genes is necessary for the activation of Hox gene expression and leukemic transformation in MLL-AF9, MLL-AF10, and MLL-AF4 leukemias [161,162,163,165,166,167]. Inhibition of both menin and DOT1L is necessary to evict MLL-AF9 from chromatin at key tumor-promoting genes in MLL-AF9-driven leukemia cells [161]. These data suggest that the interaction between menin and DOT1L plays a key role in promoting leukemogenesis in some MLLr leukemias, though not all MLL fusion proteins recruit DOT1L [168]. (For more details on the mechanistic interplay between menin and DOT1L, please see [160].) In addition to promoting oncogenic gene expression in hematopoietic cells, these oncogenic complexes containing menin and MLL may also suppress the activation of tumor suppressor genes [169]. A recent study by Soto-Feliciano and colleagues found that the menin–MLL complex antagonized the binding of the MLL3/4-UTX complex at tumor suppressor gene loci, thereby inhibiting the activation of a tumor-suppressive transcriptional program in leukemic cells [169]. Nevertheless, menin may also have leukemogenic roles independent of its interactions with MLL [170], and further research in this area is needed.

### 3.1. Small Molecule Menin Inhibitors

Small molecule inhibitors of the menin–MLL interaction (generally referred to as menin inhibitors or MIs) have proven efficacious in reducing leukemic cell outgrowth and promoting differentiation in preclinical models of both MLLr and NPM1 mutant leukemias (reviewed in [171,172,173,174,175,176]). MIs disrupt the interaction between menin and MLL, preventing wild-type MLL and MLL fusion proteins from binding to target gene promoters and activating the expression of pro-leukemic genes such as *HOXA9*, *MEIS1*, and *PBX3* (reviewed in [171,172,173,174,175,177]). Given the efficacy of MIs in preclinical models [125,144,157,170,177,178,179,180,181,182,183,184], multiple MIs are currently undergoing clinical trials in MLLr and NPM1 mutant leukemias (reviewed in [171,172,173,174,175,177,185]), including SNDX-5613, KO-539, JNJ-75276617, DS-1594b, BMF-219, DSP-5336, and BN104 (Table 1 and Table 2). To date, these MIs have shown promising safety, efficacy, and tolerability [171,172,173,174,175,177,185].

### 3.2. Menin Inhibitor Combination Therapies

Despite the promising preliminary results of early-phase clinical trials of MIs as single-agent therapies for MLLr and NPM1 mutant leukemias, some leukemias lose response to MIs alone. Perner and colleagues reported that 38.7% of individuals receiving at least two cycles of the MI SNDX-5613 (also known as revumenib) developed resistance due to acquired somatic mutations in menin that altered the MLL binding pocket. These mutations decreased SNDX-5613’s binding affinity without disrupting the menin–MLL interaction, allowing for a return of elevated Hox gene expression [186]. Resistance has also been observed in individuals without acquired menin mutations, though the pathways responsible for resistance in these cases remain uncertain [187]. In an effort to overcome resistance, combination therapies involving MIs and multiple different agents are being explored in both preclinical studies and clinical trials.

One agent that has shown efficacy in combination with MIs is the BCL-2 inhibitor venetoclax, which is used to treat AML in individuals 75 years and older [188]. Venetoclax synergizes with SNDX-5613, VTP50469 (a closely related analog of SNDX-5613, sometimes referred to as SNDX-50469), KO-539, and MI-503 in MLLr and NPM1 mutant AML cells, with the combinations more significantly reducing cell viability than any of the three MIs alone [189,190,191,192]. Importantly, the combination of venetoclax and VTP50469 primarily targets leukemic cells and has minimal impact on the viability of normal CD34+ progenitors [189]. Mechanistically, the combination of venetoclax and MIs may more significantly reduce leukemic cell viability through the repression of *HIF1A* and *HDAC9* transcription [190]. Ling and colleagues found that HDAC9 expression was necessary for MLLr AML cell proliferation and that the combination of venetoclax and MI-503 repressed HIF1α expression, which in turn inhibited the transcription of *HDAC9* [190]. In addition to venetoclax and MIs, FLT3 inhibitors have also exhibited synergy with MIs in MLLr and NPM1 mutant leukemias [193,194]. In fact, adding the FLT3 inhibitor gilteritinib to the combination of VTP50469 and venetoclax more significantly reduces NPM1/FLT3 mutant leukemic cell viability than dual therapy with venetoclax and VTP50469 [195]. Mechanistically, the three inhibitors in combination more significantly inhibit HOXA9, MEIS1, and BCL2 expression and reduce pFLT3 levels [195], which are elevated in patient-derived MLLr and NPM1/FLT3 mutant AML cells that develop resistance to the combination of VTP50469 and venetoclax treatment [192]. Due to the promising results of combining menin inhibitors with venetoclax and FLT3 inhibitors in preclinical models, venetoclax and FLT3 inhibitors are currently in clinical trials with MIs (Table 2).

In addition to BCL-2 and FLT3 inhibitors, inhibitors of multiple epigenetic proteins have also demonstrated synergy with MIs. Combining the HDAC inhibitor chidamide and MI-3 induces apoptosis and inhibits cell cycle progression in MLLr leukemias more strongly than MI-3 alone [196]. Additionally, combining the BRG1/BRM ATPase inhibitor FHD-286 with VTP50469 reduces AML burden and improves survival in both MLL1r and NPM1 mutant patient-derived xenograft models than MI alone [197]. DOT1L inhibitors have also been found to synergize with MI-2-2, MI-503, and VTP50469 in MLLr and NPM1 mutant leukemias [180,198,199,200], with combination treatment resulting in a more significant increase in differentiation compared to MI alone [180,198]. In addition, targeting epigenetic regulators such as bromodomain (BET) proteins, the lysine-specific histone demethylase 1 (LSD1), and the transcriptional coactivator CBP/p300 in combination with SNDX-5613 has also been shown to more significantly reduce MLLr and NPM1 mutant leukemic cell viability than SNDX-5613 alone [201]. Furthermore, combining VTP50469 with mezigdomide, which degrades the transcription factor Ikaros (IKZF), synergistically increases apoptosis, differentiation, and cell cycle arrest in MLLr and NPM1 mutant AML cells compared to MI alone [202,203].

Additionally, the CDK6 inhibitor abemaciclib synergizes with SNDX-5613 and KO-539 in MLLr and NPM1 mutant AML cell lines [189,191], with the combination resulting in a greater reduction in cell viability compared to menin inhibition alone [189]. Pharmacologic inhibition of the catalytic immunoproteasome subunit PSMB8 also synergizes with MI-503 to more significantly reduce leukemia cell proliferation and gene expression in MLLr and NPM1 mutant AML cell lines and patient-derived MLLr xenograft models through increased expression of the tumor suppressive transcription factor BASP1 [204]. Notably, the combination of PSMB8 and the MI SNDX-5613 inhibits the outgrowth of MI-resistant clones compared to treatment with SNDX-5613 alone [204]. Furthermore, inhibiting the RNA binding protein IGF2BP3 may also increase the efficacy of MIs in MLLr leukemias, as knocking down IGF2BP3 enhances the differentiation of MLL-AF4 leukemia cells following treatment with MI-503 [205]. In addition to these inhibitors, the RARα agonist tamibarotene synergizes with SNDX-5613 in MLLr AML cells, resulting in larger increases in apoptosis and differentiation compared to SNDX-5613 alone [206]. Overall, there are multiple promising combinatorial approaches that may combat MI resistance and improve the response of MLLr and NPM1 mutant leukemias to MIs.

### 3.3. Menin in Other Leukemias

In addition to its established oncogenic roles in MLLr and NPM1 mutant leukemias, menin may also play a role in NUP98 fusion and UBTF tandem duplication (UBTF-TD) acute leukemias. Heikamp and colleagues found that inhibiting the menin–MLL interaction in NUP98 fusion leukemia cell lines with VTP50469 promotes differentiation and inhibits leukemogenesis by evicting MLL and NUP98 fusion proteins from the chromatin at key pro-leukemic gene loci [207], suggesting that the menin–MLL interaction may also function as a key oncogene in NUP98 fusion leukemias. Additionally, Barajas and colleagues found that inhibiting menin with SNDX-5613 in primary cells derived from UBTF-TD acute myeloid leukemias inhibits leukemic cell growth and promotes differentiation. The observed anti-leukemic activity of SNDX-5613 in UBTF-TD leukemia cells is likely due to the repression of pro-leukemic gene expression, as UBTF-TDs colocalize with menin and MLL at multiple genomic loci, including those implicated in leukemia [208]. Further studies are warranted to better understand the role of menin in these leukemia subtypes and other hematological malignancies.

## 4. Digestive Cancers

### 4.1. Hepatocellular Carcinoma

While there is some disagreement in the literature as to how menin expression changes in hepatocellular carcinoma (HCC) [209,210,211,212], multiple studies report that menin expression is higher in HCC compared to non-malignant liver tissue [209,211,212]. In HCC, menin expression is positively correlated with fibrosis grade and tumor size [209]. Furthermore, high menin expression is associated with more aggressive tumor phenotypes, higher expression of α-fetoprotein (AFP), and lower overall and tumor-free survival [211,212]. In HCC cell lines, both pharmacologic and genetic inhibition of menin reduces HCC cell and tumor xenograft growth [211,212,213]. Furthermore, overexpressing menin in HCC cell lines increases the expression of the pro-inflammatory cytokine IL-6, which has been shown to play a key role in the initiation and progression of HCC [214]. Together, these data indicate that menin may promote HCC tumorigenesis.

#### 4.1.1. Oncogenic Gene Expression and Signaling Pathways

Mechanistically, menin affects multiple different pathways to promote HCC development and progression. A study by Xu and colleagues found that menin’s pro-proliferative activity in HCC was dependent on YAP1 [211], a known driver of HCC [215,216,217]. Menin promoted the expression of YAP1 through increased H3K4 methylation at the *YAP1* promoter [211,212]. Knocking down YAP1 expression in HCC cells completely abolished the increase in colony formation and IL-6 expression observed with menin overexpression [211]. Menin’s effect on YAP1 expression was seemingly limited to HCC, as knocking down or overexpressing menin in breast, lung, and Wilms tumor cell lines had no effect on YAP1 [211]. Interestingly, HCCs upregulate the expression of menin–MLL complex members such as WDR5, RBBP5, ASH2L, and MLL, in addition to menin [211], suggesting that menin promotes HCC pathogenesis by increasing the activity of the menin–MLL complex and subsequently increasing YAP1 activity.

In addition to regulating YAP1 expression, there is some evidence that menin also promotes the expression of PEG10, which functions as an oncogene in HCC [212,218,219]. Menin may also promote HCC tumorigenesis through the dysregulation of Hox gene expression, similar to its function in leukemia. Xu and colleagues found that menin localizes to the promoters of *HOXA7*, *HOXA9*, and *HOXA13* and that menin overexpression increases both their expression and H3K4 promoter methylation [211].

Menin may further promote HCC tumorigenesis by exacerbating Wnt activation in HCCs with activating Wnt mutations, as the small molecule menin inhibitor MI-503 was also able to reverse the activated *CTNNB1*-associated gene expression signature in an HCC cell line with an activating mutation in *CTNNB1* [212]. In addition, Xu and colleagues found that inactivating one allele of menin in a mouse model of liver tumorigenesis reduced activation of the Akt, Stat3, and mitogen-activated protein kinase (MAPK) pathways [211]. Together, these data indicate that menin promotes HCC development and progression in part by activating various transcriptional programs and signaling cascades that promote cell growth, survival, migration, and oncogenic transformation.

#### 4.1.2. IGF Signaling

An additional mechanism by which menin promotes HCC tumorigenesis is through the upregulation of the classically imprinted gene *IGF2*. Zheng and colleagues found that menin increases IGF2 expression in HCC [220]. In some HCCs, IGF2 expression and IGF1R-Akt pathway are upregulated [220]. Menin increases IGF2 expression in HCC cell lines by reducing DNA methylation and increasing H3K4me3 marks at the *IGF2* promoter [220]. Menin also increases activating H3K79 methylation, decreases repressive H3K27 methylation, and increases the localization of DOT1L and MLL to the *IGF2* promoter [220]. The IGF2 pathway plays a key role in HCC development and progression, activating Ras, MAPK, and PI3K/Akt signaling downstream of IGF1R activation by IGF1/2 [221]. Inhibiting the menin–MLL interaction with MI-3 repressed the growth of HCC cell lines that highly express IGF2 by reducing H3K4me3 at the IGF2 promoter, decreasing IGF2 expression, and subsequently inhibiting IGF1R-Akt pathway activation [220]. Menin can, therefore, promote HCC tumorigenesis by coordinating the epigenetic regulation of IGF2 in the liver, leading to the activation of pro-proliferative and pro-survival signaling in the liver.

#### 4.1.3. The Extracellular Matrix and Fibrosis

Liver fibrosis is a key feature of cirrhosis, which is a major risk factor for HCC [222,223]. Liver fibrosis results from the increased deposition of extracellular matrix components like collagen in the liver [224,225]. Menin expression is positively correlated with fibrosis status [226]. Furthermore, menin regulates the expression of hepatic collagen α2(I), and menin expression is positively correlated with collagen α2(I) expression in HCC [209]. Knocking down menin in HSC-derived cell lines significantly reduces collagen α2(I) expression, and collagen α2(I) expression is partially restored by activating TGFβ signaling [209], a known activator of collagen α2(I) gene transcription [227], in a manner independent of the TGFβ-mediated fibrogenic response mediator SMAD3 [209,228,229]. Menin may also play a role in hepatic stellate cell (HSC) activation during fibrogenesis, as activated HSCs in culture have higher menin expression compared to freshly cultured hepatocytes and Kupffer cells [209].

#### 4.1.4. Therapeutic Potential of Menin Inhibitors for HCC

While MIs alone have been shown to reduce HCC growth, multiple studies suggest that combining MIs with other inhibitors can more effectively inhibit HCC progression [212,213]. Combining an MI with an EZH2 inhibitor or the multi-tyrosine kinase inhibitor sorafenib had a greater inhibitory effect on HCC than MI alone [212,213]. Menin inhibitors may, therefore, be a potential therapeutic strategy for HCC when used in combination with other inhibitors, some of which are already in clinical use.

#### 4.1.5. Menin as a Tumor Suppressor in HCC

While the literature generally supports menin functioning as a tumor promoter in HCC, there are some conflicting data suggesting that menin can alternatively play a tumor-suppressive role in HCC. Gang and colleagues found that menin expression was lower in HCC tumor tissue compared to non-malignant liver tissue and that menin expression was lower in multiple HCC cell lines compared to a non-malignant liver cell line [210]. In addition, overexpressing menin in HCC cells reduced proliferation while knocking down menin increased proliferation [210]. Menin overexpression also reduced the expression of the pro-inflammatory cytokines IL-1β and Cox-2 following stimulation with TNFα [210]. Mechanistically, menin inhibited cell proliferation through inhibition of NF-κB signaling, as the expression of a dominant-negative version of the NF-κB inhibitor IκBα completely abolished the increase in proliferation observed with menin silencing [210]. Additionally, menin regulated pro-inflammatory cytokine expression at least partially through the recruitment of the histone deacetylase Sirt1 to the *TNFA*, *IL1B*, and *COX2* promoters [210]. Menin also recruited Sirt1 to deacetylate the NF-κB subunit p65, as acetylated p65 protein levels decreased with increasing menin expression and increased upon menin knockdown by siRNA [210]. NF-κB plays a tumor-promoting role in HCC, increasing cell proliferation, invasion, and inflammation while inhibiting apoptosis [230], suggesting menin may suppress liver tumorigenesis in some contexts by downregulating NF-κB.

The conflicting data regarding whether menin promotes or suppresses HCC tumorigenesis indicate that additional studies are needed to better establish the tumor-promoting and tumor-suppressive functions of menin in HCC. Furthermore, additional investigation is warranted to identify the contexts and mechanisms by which menin can switch from a tumor promoter to a tumor suppressor and vice versa in HCC.

### 4.2. Cholangiocarcinoma

There is limited data on menin’s function in cholangiocarcinoma, though the existing data suggest that menin may have tumor-suppressive roles. Menin expression is inversely correlated with tumor grade in a cholangiocarcinoma cohort [226], and high menin expression is associated with improved survival in another [226]. Menin expression is also lower in multiple cholangiocarcinoma cell lines compared to a nonmalignant human biliary epithelial cell line [231], suggesting that menin may be lost during cholangiocarcinogenesis. One mechanism for this loss of menin may be increased activity of miR-24, which is upregulated and associated with a worse prognosis in multiple gastrointestinal cancers [232,233,234,235]. Similar to its role in pancreatic and parathyroid NETs, miR-24 targets MEN1 transcripts for degradation in the biliary tract [231] and forms a negative feedback loop with menin that regulates the expression of cell cycle and apoptotic genes [81,82]. In cholangiocarcinoma, inhibiting miR-24 increases menin expression, increases fibrosis, decreases pro-angiogenic gene expression, and decreases tumor xenograft growth [231], consistent with its role as a negative regulator of menin.

In addition to suppressing cholangiocarcinoma cell proliferation, migration, and invasion [231], menin also suppresses the expression of pro-angiogenic factors such as VEGFA/C, VEGFR2/3, angiopoietins-1/2, and angiopoietin receptors TIE1/2 in a cholangiocarcinoma cell line and tumor xenograft model [231]. This suppression of pro-angiogenic gene expression may occur through the known menin-binding partner JunD [52,236], which has previously been shown to reduce tumor angiogenesis in subcutaneous tumors formed by Kras mutant MEFs [237]. However, whether there is any meaningful interaction between JunD and menin in cholangiocarcinoma remains to be explored. Future studies are therefore needed to characterize the menin–JunD interaction in the biliary epithelium and further explore its role in cholangiocarcinogenesis.

### 4.3. Pancreatic Ductal Adenocarcinoma

Menin expression is downregulated in pancreatic ductal adenocarcinomas (PDACs) compared to surrounding non-malignant pancreatic tissue [238,239]. Menin expression decreases with increasing PDAC stage [239], and individuals whose tumors highly express menin tend to have improved survival compared to those with low tumor menin expression [240]. Together, these data suggest that loss of menin may contribute to PDAC progression.

Menin decreases proliferation and tumor xenograft growth in various PDAC cell lines by inhibiting cell cycle progression [238,239,240], similar to its role in pNETs. Mechanistically, menin binds to the DNA methyltransferase DNMT1 and reduces its localization at the promoters of the cyclin-dependent kinase (CDK) inhibitors p18 and p27, thereby promoting p18 and p27 expression [239].

Menin also inhibits PDAC cell proliferation through the inhibition of pro-proliferative Hedgehog signaling [239]. Overexpressing menin in PDAC cell lines reduces the expression of Hedgehog target genes *GLI1* and *GAS1* and prevents the pro-proliferative effects of the Hedgehog agonist SAG [239]. Menin has been previously shown to influence Hedgehog signaling in pNETs that develop in the context of MEN1 syndrome [49]. Interestingly, Hedgehog signaling may, in turn, regulate menin expression in PDAC through changes in menin promoter methylation by DNMT1 [239]. Inhibiting the Hedgehog pathway increases menin expression, while activating Hedgehog signaling decreases menin expression in a DNMT1-dependent manner [239].

However, while menin inhibits cell proliferation in PDAC, it can also promote epithelial-to-mesenchymal transition (EMT) [240]. PDAC cell lines overexpressing menin exhibit a more mesenchymal morphology and gene expression profile compared to isogenic control lines [240]. Furthermore, menin overexpression increases motility and migration gene signatures in PDAC cell lines, indicating that menin may also promote PDAC metastasis. Treating PDAC cells overexpressing menin with an HDAC inhibitor blocks the downregulation of epithelial gene expression, suggesting that menin promotes EMT in PDAC through deacetylation of epithelial gene promoters [240]. Additionally, the switch from an epithelial-like state to a mesenchymal-like state upon menin overexpression can be recapitulated by activating TGFβ signaling [240], which is a known promoter of cancer cell invasion, metastasis, and EMT [241,242]. While treating cells with exogenous TGFβ does not affect menin expression, menin overexpression upregulates TGFβ signaling [240], suggesting that menin promotes EMT in PDAC by upregulating TGFβ signaling.

Whether menin functions as an inhibitor of PDAC cell growth or a promoter of EMT in PDAC may depend on the transcription factor C/EBPβ [240]. Overexpressing the C/EBPβ isoform LAP2 in cells overexpressing menin results in a more significant reduction in PDAC cell proliferation and tumor xenograft growth than menin overexpression alone [240]. Overexpression of LAP2 also rescues epithelial gene expression and returns PDAC cells to a more epithelial morphology [240], suggesting that C/EBPβ expression suppresses PDAC cell growth, while loss of C/EBPβ expression promotes EMT. Interestingly, while C/EBPβ regulates whether menin’s activity in PDAC is tumor-suppressive or oncogenic, menin can, in turn, suppress the expression of C/EBPβ through recruitment of HDACs to the *CEBPB* promoter [240]. Menin, therefore, plays a dual role in PDAC, suppressing cell growth when C/EBPβ is present and promoting metastasis when it is not.

### 4.4. Gastric Adenocarcinoma

While the role of menin as a tumor suppressor in gastric NETs is well-established, menin’s role in gastric adenocarcinoma, hereafter referred to as gastric cancer, is not well-defined. One study reported that menin protein expression was lower in gastric cancer compared to surrounding benign gastric tissue [243]. Individuals with low menin-expressing gastric cancers were more likely to have metastatic disease compared to those with high menin expression, further suggesting that menin plays a tumor-suppressive role in gastric cancer. In line with these clinical findings, overexpressing menin in a gastric adenocarcinoma-derived cell line decreased cell growth, at least partially by decreasing PI3K-Akt and NF-κB signaling [243]. Notably, menin has previously been reported to suppress NF-κB and PI3K-Akt signaling pathways in parathyroid tumors [244] and MEFs [15], indicating that menin regulates cell growth in multiple tissues through PI3K-Akt and NF-κB signaling. In gastric adenocarcinoma-derived cell lines, menin expression was also found to downregulate the expression of IQ motif containing GTPase-activating protein 1 (IQGAP1) [243]. IQGAP1 is a scaffold protein that plays a role in multiple cellular processes, including cell–cell adhesion [245], cell migration [246], transcription [247], and signal transduction [248]. Interestingly, IQGAP1 has been previously suggested to play a tumor-promoting role in multiple cancers, including gastric cancer [247,249,250,251,252], suggesting that menin may also help suppress gastric cell proliferation through the repression of *IQGAP1*. However, further research is needed to better characterize the relationship between menin and IQGAP1. Additional mechanistic studies in more clinically relevant models are also necessary to better understand the role of menin in gastric cancer.

### 4.5. Colorectal Cancer

Menin is overexpressed in colorectal cancer (CRC) compared to benign colonic tissue; however, menin alone does not affect CRC growth [253]. Nonetheless, menin has been shown to be involved in multiple pathways that are important for resistance to small molecule EGFR inhibitors (iEGFRs) in CRC. One of these pathways involves the regulation of SKP2 expression [253]. SKP2 is an oncogene known to promote CRC growth [254,255,256]. Menin binds to the promoter of *SKP2* and drives SKP2 expression, with inhibition of menin leading to a reduction in expression of SKP2. Additionally, menin promotes SKP2 expression, at least partially, by increasing H3K4me3 deposition and RNA polymerase II localization at the *SKP2* promoter [253]. It has also been demonstrated that the reduction in SKP2 by inhibiting menin, either genetically or pharmacologically, synergizes with iEGFRs to reduce CRC cell growth, increase CRC cell death, and reduce tumor xenograft growth more effectively than treatment with iEGFRs alone [253].

Menin was also found to negatively regulate glycolysis in CRC cells in a manner independent of mTOR [257]. Inhibiting menin through multiple methods led to an increase in glycolysis, and this increase in glycolysis also increased CRC cell sensitivity to iEGFRs, suggesting an additional mechanism whereby menin promotes CRC resistance to iEGFRs by repressing glycolysis [257]. Treating CRC cells with an autophagy inhibitor further increased their sensitivity to combined treatment with EGFR and MIs, indicating that autophagy induction is an important mechanism protecting CRC cells against combined iEGFR/MI treatment [257].

Menin also promotes CRC cell survival in lipid-poor conditions [258]. Inhibiting menin increases the transcription of LXR-regulated genes *ABCA1* and *ABCG1*, which encode cholesterol exporters, leading to decreased intracellular cholesterol levels in CRC cell lines [258]. Interestingly, lower levels of intracellular cholesterol are also observed in the benign colonic epithelium of mouse models lacking menin expression in the colonic epithelium. Inhibiting menin significantly decreases cell growth and increases cell death under lipid-poor conditions. This decrease in cell growth is at least partially due to increased cholesterol efflux from CRC cells as a result of increased ABCA1 and ABCG1 expression following menin inhibition, as the phenotype can be recapitulated by treating CRC cells with an LXR-agonist [258]. Notably, menin inhibitors BMF-219 and SNDX-5613 are currently in clinical trials for CRC (Table 3).

Menin may also inhibit telomerase activity in the colon, and this negative regulation of telomerase may be dysregulated in CRC [259]. A study by Ao and colleagues found that menin binds to the promoter of telomerase reverse transcriptase (TERT; the catalytic component of telomerase) and negatively regulates TERT expression in CRC cells [259]. Correspondingly, menin was downregulated in some CRCs compared to adjacent non-malignant colon tissue, while TERT was upregulated [259]. Overexpressing menin in CRC cells reduced TERT expression and decreased cell viability [259]. Interestingly, the increase in TERT was due to increased ubiquitylation and subsequent proteasomal degradation of menin by the cullin–RING ubiquitin ligase 4 (CRL4) complex [259], which is upregulated in pro-inflammatory microenvironments, such as those that may occur in the context of CRC [260]. Telomerase is aberrantly activated in cancers, including CRC, and telomerase activation has been shown to increase the proliferative potential of cancer cells (reviewed in [261,262,263]). Therefore, menin may suppress CRC tumorigenesis in some contexts by reducing the proliferative potential of CRC by inhibiting telomerase activity [259]. Nevertheless, when the authors examined telomere lengths, CRC tumors were found to have significantly shorter telomeres compared to adjacent non-malignant tissue despite the increase in TERT expression [259]. There are similar conflicting findings in other cell types, in which menin suppressed TERT transcription in some contexts yet had no apparent effect on telomerase activity [264,265,266]. Together, these data suggest that while loss of menin may increase TERT expression in the colon, this increase in TERT may not translate to increased telomerase activity, telomere lengthening, or subsequent tumorigenicity of the colon. Further study is needed to better understand the functional significance of elevated TERT transcription in the colon.

## 5. Breast and Gynecological Cancers

### 5.1. Breast Cancer

The role of menin in breast cancer is highly context-dependent. Some data suggest that menin has a dual role in breast cancer, with menin playing a tumor suppressive role in the non-malignant mammary epithelium [267,268,269] and a tumor promoting role in sporadic estrogen receptor (ER)-positive and ER-negative breast cancers [267,268,269,270,271,272,273,274,275,276]. However, other data suggest that menin also plays a tumor-suppressive role in ER-positive breast cancers [268,269,270,277,278]. Therefore, the specific conditions and mechanisms of tumor suppression versus oncogenesis remain active areas of investigation.

#### 5.1.1. Menin as a Tumor Suppressor

In primary luminal progenitor cells of the breast ductal epithelium, menin suppresses proliferation and downregulates multiple extracellular matrix genes associated with invasive breast cancer, including the matrix metalloproteinases MMP9 and MMP3 [267]. Notably, the genes dysregulated by the loss of menin in these primary ductal epithelial cells differ from those dysregulated in luminal breast cancer cells [267], suggesting that menin regulates non-malignant and malignant breast tissue differently. Interestingly, clinical data in individuals with MEN1 syndrome suggest that MEN1 patients may be at increased risk of developing breast cancer (reviewed in [279]). Notably, a study by Dreijerink and colleagues found that individuals with MEN1 syndrome were more likely to develop breast cancer and were diagnosed with ductal breast cancers at an earlier age than individuals without MEN1 syndrome [280]. This phenotype has also been observed in *Men1* mutant mice [268,269], where menin knockout in breast ductal epithelial cells substantially increases the incidence of mammary intraepithelial neoplasia (MIN) [268]. Further supporting a protective role of menin in the breast, menin expression is reduced in a large proportion of sporadic breast carcinomas outside the context of MEN1 syndrome [268]. Furthermore, in ER-positive breast cancers, specifically, lower menin expression is associated with larger tumors, higher tumor grades, and worse survival [269].

Mechanistically, menin may suppress breast tumorigenesis by binding to the p65 subunit of the NF-κB complex and suppressing NF-κB-regulated gene expression [277]. NF-κB promotes the expression of multiple genes that contribute to tumor progression, such as cyclin D1, c-Myc, BCL2, snail, vimentin, and MMP2/3/9 [277]. Menin has previously been shown to exert its tumor-suppression function by interacting with NF-κB and repressing NF-κB-mediated transcriptional activation [281]. Importantly, the heterotrimeric replication protein A (RPA) complex component RPA2, which is overexpressed in breast cancer [277,282], can outcompete NF-κB for binding to menin [277]. The resulting loss of NF-κB sequestration has been shown to increase cell proliferation, adhesion, migration, invasion, and EMT gene expression [277].

Menin may also suppress ER-positive breast tumorigenesis through inhibition of the PI3K/Akt/mTOR signaling pathway [270], similar to its role in pNETs. The mTORC1 pathway has previously been shown to promote breast cancer cell growth and play a key role in breast cancer development and resistance to endocrine therapy [283]. In ER-positive luminal breast cancer cell lines, menin knockdown increases mTORC1 activation, likely through the activation of Akt, and subsequently increases the formation of the eukaryotic initiation factor 4F (eIF4F) complex [270]. eIF4F is one of the downstream targets of the mTORC1 pathway, and it plays a key role in the initiation of cap-dependent protein translation [284], which is frequently dysregulated during tumorigenesis (reviewed in [285,286]). Menin’s suppression of mTOR signaling is also supported by clinical data, as individuals with low menin-expressing tumors receiving tamoxifen were found to benefit from the addition of the mTOR inhibitor everolimus, while individuals with high menin-expressing tumors receiving tamoxifen saw no additional survival benefit from adding everolimus, suggesting that menin was already suppressing mTOR [270].

#### 5.1.2. Menin as a Tumor Promoter in ER-Positive Breast Cancers

While the above data demonstrate that menin can suppress breast tumorigenesis, other contradictory data indicate that menin can promote tumorigenesis in the breast. A study by Massey et al. found that menin expression was higher in breast tumor tissue than in adjacent non-malignant tissue due to decreased DNA methylation at the *MEN1* promoter [273]. Furthermore, menin expression was higher in breast tumors that had metastasized to the lymph nodes [273] compared to non-metastatic tumors, and high menin expression was associated with worse survival [268,273].

Mechanistically, menin may promote ER-positive breast tumorigenesis through its interaction with ERα. While it is unclear whether menin regulates the expression of ERα itself in breast cancer [267,268,269,276], menin directly interacts with ERα in a hormone-dependent manner through the AF2 domain of ERα [271,275,276]. Through this interaction, menin promotes increased H3K4 trimethylation at ERα target gene promoters, thereby increasing the transcription of ERα target genes [267,276], including the estrogen-responsive oncogene *MYC* [270,287,288,289]. Estrogen and ERα have been shown to promote the growth of breast cancer cells [267,290,291,292], and Myc is known to play a key role in proliferation, metabolism, and apoptosis in various cancers, including breast cancer [288,293,294]. Consistent with its upregulation of Myc [295], menin expression resulted in cell cycle activation through the upregulation of genes encoding cyclins A2/B/E/D1, CDK2/4, and p53 [270] and downregulation of genes encoding the cell cycle inhibitors p27 and p21 [270]. Accordingly, inhibiting menin in ER-positive luminal breast cancer cell lines reduced cell growth [267,269,270], suggesting that menin promotes breast cancer progression by augmenting oncogenic ERα target gene expression downstream of estrogen signaling. In this way, menin may confer resistance to selective estrogen receptor modulators [271,272], which are a standard of care for patients with hormone receptor-positive breast cancers [296], especially in patients with high tumor expression of menin [271,272].

Menin may also promote breast tumorigenesis through the regulation of transcription. Menin interacts with and promotes the expression of the transcription factors FOXA1 and GATA3 in ER-positive luminal breast cancer cells [269]. GATA3 is expressed in some breast cancers and functions as a coregulator of *ESR1* (encoding ERα) gene transcription [297]. FOXA1 functions as a coregulator of ERα and plays an important role in mammary cell differentiation and breast tumorigenesis [298]. Menin is present at enhancers bound by FOXA1 and GATA3, and menin increases the transcription of multiple genes regulated by these enhancers in ER-positive luminal breast cancer cell lines [267]. Menin also complexes with the H3K79 histone methyltransferase DOT1L and the epigenetic reader protein BAZ1B in ER-positive luminal breast cancer cells [275]. Menin, DOT1L, and BAZ1B are overexpressed in ductal breast tumors compared to non-malignant mammary tissue [275], and they colocalize to both distal and proximal enhancer-like sites for multiple genes involved in pathways known to play a role in breast cancer, such as estrogen signaling, p53 signaling, HIF1α signaling, death receptor signaling, PI3K/Akt/mTOR signaling, Myc signaling, cell cycle regulation, and EMT [275]. Simultaneous inhibition of menin, DOT1L, and BAZ1B synergistically slows the proliferation of both antiestrogen-sensitive and -resistant ER-positive cell lines [275], further supporting menin’s function as a tumor promoter through its association with epigenetic modifiers. This suggests that targeting menin, DOT1L, and BAZB1 in combination may be a potential therapeutic strategy for treating ER-positive breast cancers.

#### 5.1.3. Menin as a Tumor Promoter in ER-Negative Breast Cancers

While the role of menin in the context of ER-positive breast cancers has been heavily studied, the role of menin in ER-negative breast cancers is less understood. Menin expression is more heterogeneous in triple-negative breast cancers (TNBC) [274], with a trend towards lower expression [269,274] compared to ER-positive breast cancers. Despite this heterogeneity, however, menin seems to function as a tumor promoter in ER-negative breast cancers as well. Inhibiting menin in menin-expressing TNBC cells decreases cell proliferation and migration [299], increases apoptosis [274], delays tumor growth [274,299] and lung metastasis [299], improves survival [299], and increases sensitivity to multiple chemotherapeutic agents [274,299]. These data suggest that menin inhibition may be a potential therapeutic strategy for improving response to chemotherapy in TNBC patients with high tumor menin expression [274,299].

Mechanistically, menin promotes TNBC proliferation and migration through its interaction with MLL [274,299]. Specifically, pharmacologically inhibiting the menin–MLL complex upregulates antiproliferative genes involved in the p53 pathway and DNA repair and downregulates multiple pro-proliferative genes, including genes involved in cell division and cell cycle progression, MAPK signaling, PI3K-Akt-mTOR signaling, and KRAS signaling [299]. Menin–MLL also promotes the secretion of cytokines IL-6, IL-8, and TGFβ1 by TNBC cells, which promotes migration through the activation of actin filament assembly via the IL-6/8/pSTAT3/Arp3 axis and activation of myosin contractility via the TGFβ1/Gli2/ROCK1/2/pMLC2 axis [299]. Interestingly, unlike ER-positive breast cancers, menin has no effect on Myc expression in TNBC [270], suggesting that menin regulates breast tumorigenesis through different mechanisms depending on the ER status of the tumor. Analysis of menin’s other binding partners in TNBC suggested that menin may also regulate mRNA 3′-end processing [274], which is dysregulated in some cancers (reviewed in [300]), suggesting menin may also promote hormone receptor-negative breast tumorigenesis through dysregulation of mRNA processing. Nevertheless, the fact that menin can also function as a tumor suppressor indicates that menin alone is neither necessary nor sufficient for breast cancer development. This highlights the need for additional studies to better understand the specific conditions determining whether menin promotes or suppresses breast tumorigenesis.

### 5.2. Ovarian Cancer

Compared to breast cancer, the role of menin in gynecological malignancies is less defined. To date, there has only been one study focusing on the role of menin in ovarian cancer. This study found that menin was expressed more highly in ovarian cancer compared to benign ovarian tissue and suggested that menin may function as a tumor promoter in ovarian cancer [301]. Inhibiting menin, either genetically or pharmacologically, in multiple ovarian cancer cell lines reduced proliferation. This antiproliferative effect was more pronounced in metastatic cell lines, with menin expression being necessary for their survival [301]. Inhibiting menin in ovarian cancer cell lines led to the upregulation of genes involved in integrin signaling and the downregulation of genes associated with cell cycle regulatory pathways, aryl hydrocarbon receptor signaling, Myc signaling, and KRAS signaling [301]. Menin expression was also found to be correlated with the expression of the H3K79 histone methyltransferase DOT1L in human ovarian cancers [301], similar to MLL-rearranged leukemia and endocrine-resistant breast cancer cells, where menin regulates DOT1L activity in a pro-tumorigenic manner [198,302]. Combination treatment of both chemotherapy-sensitive and chemotherapy-refractory ovarian cancer cell lines with inhibitors of menin and DOT1L further reduced proliferation compared to treatment with either inhibitor alone [301]. Taken together, these findings suggest that menin, at least in part through cooperation with DOT1L, promotes ovarian cancer growth through the regulation of genes involved in multiple key cellular pathways that regulate cell proliferation and survival.

### 5.3. Endometrial Cancer

There is similarly limited data examining the role of menin in endometrial cancer. Menin was found to be upregulated in endometrial cancers, and high tumor menin expression was associated with worse relapse-free and overall survival [303]. Also, in an unbiased drug screen of epigenetic regulators in mouse-derived endometrial cancer organoids, menin inhibitors MI-136 and MI-463 were found to significantly reduce organoid growth in vitro [303]. MI-136 also reduced organoid growth in an orthotopic xenograft mouse model and the growth of patient-derived endometrial tumoroids in vitro [303]. Genetically depleting menin using CRISPR/Cas9 in menin-expressing endometrial cancer-derived organoids similarly reduced tumor organoid growth [303].

Mechanistically, inhibiting menin reduced cell proliferation and reduced the expression of hypoxia-inducible factor (HIF) target genes. Menin expression was highly correlated with HIF1A expression in human endometrial cancer samples. Knocking out *Hif1a* or *Hif1b* in mouse-derived endometrial tumoroids partially recapitulated the effect of menin inhibition by MI-136 or genetic knockout on tumoroid growth [303], suggesting that menin may promote endometrial cancer growth through the activation of HIF target genes. Inhibiting additional members of the menin–MLL complex, MLL1 and Ash2L, similarly reduced tumor organoid growth and HIF target gene expression [303], suggesting the menin–MLL complex functions as a tumor promoter in endometrial cancer.

To date, this is the only study to investigate the role of menin in endometrial cancer. Additional studies are, therefore, needed to corroborate the oncogenic role of menin in endometrial cancer and to further explore the mechanisms by which menin regulates endometrial tumorigenesis.

## 6. Other Cancers

### 6.1. Prostate Cancer

Menin is overexpressed in prostate cancer compared to non-malignant prostate tissue, with menin expression being highest in metastatic prostate cancers [304,305,306,307]. Menin expression is also higher in castration-resistant prostate cancer compared to hormone-sensitive prostate cancer [304,305,306]. Additionally, high menin expression in prostate cancer is associated with decreased survival [304,305,306] and may contribute to taxane resistance in castration-resistant prostate cancer [308].

#### 6.1.1. AR-Positive Prostate Cancer

Menin promotes the growth of androgen receptor (AR)-positive prostate cancer cell lines [304,306,307,309,310], with multiple groups showing that knocking down menin in human AR-positive prostate cancer cell lines reduces proliferation and tumor xenograft growth [304,306,307,309]. In addition, inhibiting menin with MI-503 reduces cell proliferation, colony formation, and tumor xenograft growth in AR-positive cell lines [304,306,310].

Mechanistically, menin promotes AR-positive prostate cancer growth through the activation of AR signaling [304,307,309]. The menin–MLL complex is a co-activator of AR signaling in prostate cancer, with menin recognizing the N-terminal domain of AR [304]. Knocking down menin expression and inhibiting the menin–MLL complex with either MI-503 or MI-136 in AR-positive cell lines reduces AR [307] and AR target gene [304,307,309] expression. Menin also binds to the proximal *AR* promoter and regulates *AR* transcription via H3K4 trimethylation [307], and inhibiting menin reduces the recruitment of both the menin–MLL complex and AR to AR target genes but has no effect on the interaction between menin and AR [304,307]. Interestingly, knocking down MLL and the menin–MLL complex member ASH2L behaves similarly to menin knockdown in reducing AR-target gene expression, cell proliferation, and tumor growth [304]. The effect of menin knockdown on AR-positive cell line proliferation can also be recapitulated by knocking down AR [307], and the menin inhibitor MI-136 inhibits AR-positive cell proliferation in a manner similar to the second-generation FDA-approved anti-androgen MDV-3100 [304]. Interestingly, prostate cancer cells with low AR expression have higher expression of the stem cell marker CD44, and knocking down menin in prostate cancer cells similarly leads to increased expression of CD44 [307], suggesting that menin may promote the dedifferentiation of prostate cancer cells by decreasing AR expression. These data indicate that menin promotes prostate cancer tumorigenesis by increasing AR expression and activation through its functions as a member of the menin–MLL complex.

Menin also promotes AR-positive prostate cancer growth by activating Myc. Myc expression is associated with increased cell proliferation in AR-positive cell lines [309], and menin promotes Myc expression in AR-positive cells through the methylation of H3K4 at the *MYC* enhancer [309,310] and by increasing the expression of the lncRNA *PCAT1* [309], which has previously been shown to promote prostate cancer pathogenesis through the regulation of Myc expression [311]. Menin’s regulation of Myc expression is at least partially controlled by AR signaling, as activation of AR promoted menin expression and its localization to the *MYC* promoter and 3′ enhancer [309]. Menin may also promote AR-positive prostate cancer growth by promoting JunD [310] and TMPRSS2 [307] expression. JunD has previously been shown to promote the proliferation of prostate cancer cells through Myc signaling [312], and inhibiting menin with either siRNA knockdown or MI-503—a menin inhibitor that blocks the menin–JunD interaction in addition to the menin–MLL interaction [313]—reduces JunD expression in AR-positive cell lines [310]. Silencing menin in AR-positive prostate cancer cell lines also reduces the expression of TMPRSS2 [307], an androgen-responsive transmembrane serine protease that promotes prostate cancer growth and metastasis [314,315,316]. Taken together, these data indicate that menin functions as a tumor promoter in AR-positive prostate cancer primarily through the activation of AR and Myc signaling.

#### 6.1.2. AR-Negative Prostate Cancers

Unlike AR-positive prostate cancers, there are conflicting data regarding whether menin promotes or suppresses prostate tumorigenesis in AR-negative prostate cancers. Multiple studies found that menin promotes the growth of AR-negative prostate cancer cell lines [304,305,306]. In these studies, knocking down menin in AR-negative cells increases apoptosis and reduces cell proliferation and tumor xenograft growth [305,306]. Inhibiting menin with MI-503 also reduces cell proliferation, colony formation, and tumor xenograft growth in AR-negative cell lines [304,306,310]. However, a study by Teinturier and colleagues found that menin knockdown had no effect on cell proliferation [307], and an additional study by Luo and colleagues found that menin knockdown increased anchorage-independent cell growth and colony formation while inhibiting menin with MI-503 increased tumor xenograft growth [310].

In addition to regulating proliferation, menin also regulates cell migration in AR-negative prostate cancer cell lines [306,310], though as with AR-negative cell growth, there are conflicting data on whether menin promotes or suppresses migration. Kim et al. found that menin promoted cell migration in AR-negative cell lines without affecting AR-positive cell line migration [306]. However, Luo et al. found that menin decreased cell migration, and AR-negative tumor xenografts displayed a more invasive phenotype in mice treated with MI-503 compared to vehicle-treated mice [310]. Mechanistically, menin was found to localize to multiple ECM-related genes [305] and to suppress the expression of multiple metastasis-related genes [306] in an AR-negative prostate cancer cell line compared to an AR-positive line. In AR-negative cells, menin downregulated the expression of metastasis-related genes by increasing H3K9me3 deposition at their promoters via the H3K9 methyltransferase SUV39H1 [306]. Additionally, Luo and colleagues found that silencing menin in AR-negative cell lines decreased the expression of the epithelial cell marker E-cadherin [310]. Silencing menin also increased the expression of Twist, a negative regulator of E-cadherin gene transcription that has been shown to promote epithelial-to-mesenchymal transition (EMT) in multiple cancers, and HIF1A, which has been shown to regulate Twist expression [310,317]. This suggests that menin plays a tumor-suppressive role by limiting the initiation of pro-EMT transcriptional programs in AR-negative prostate cancers. The conflicting data on how menin affects AR-negative prostate cancer cell proliferation and migration suggest that the role of menin in AR-negative prostate cancers may be context-dependent.

Mechanistically, it is uncertain how menin promotes the growth of AR-negative prostate cancers. However, Luo and colleagues found that menin inhibits AR-negative cell line growth by suppressing Myc expression through the repression of *MYC* transcription [310]. Loss of menin promoted loop structure formation at the *MYC* locus between the *MYC* enhancer and proximal promoter and increased the binding of JunD and β-catenin to the *MYC* enhancer and promoter, respectively [310]. Both JunD and β-catenin have previously been shown to promote Myc expression in other cancers [287,318,319,320]. In addition to increasing their localization to Myc regulatory sequences, menin suppressed the activation of JunD and β-catenin in AR-negative prostate cancer cell lines. Treating AR-negative cells with MI-503 reduced JunD expression [310], and menin knockdown increased the nuclear localization of JunD [310]. Menin knockdown and inhibition with MI-503 also increased the nuclear localization of β-catenin [310], indicating that menin suppresses activation of both the Wnt signaling pathway and JunD. Interestingly, JunD and β-catenin were necessary for menin’s tumor-suppressive activity in AR-negative cell lines [310], as knocking down JunD or β-catenin abolished the increased colony formation observed with menin knockdown [310]. JunD and β-catenin have previously been found to play an oncogenic role in prostate cancer [312,321], indicating that menin can increase the tumorigenic potential of AR-negative prostate cancers by increasing the activation of JunD and β-catenin [310]. These data also indicate that differences in Myc regulation may be at least partially responsible for menin’s switch from being a tumor promoter in AR-positive prostate cancers to a tumor suppressor in some AR-negative prostate cancers since menin has opposing effects on Myc expression in prostate cancer cell lines depending on whether AR is expressed.

#### 6.1.3. Similarities and Differences Based on AR Status

Menin may behave differently in AR-positive and AR-negative cells due to differential regulation of the expression of multiple genes in addition to *MYC* [305,306]. Menin localizes to different genes in AR-positive cells compared to AR-negative cells [305,306]. Compared to AR-positive cells, in AR-negative cells, menin is enriched at genes involved in the PI3K/Akt pathway and drug resistance pathways, including EGFR tyrosine kinase inhibitor resistance and platinum drug resistance [305]. The PI3K/Akt signaling pathway is associated with disease progression, resistance to androgen deprivation, resistance to chemotherapy, and poor survival in castration-resistant prostate cancer [322,323,324]. Menin also preferentially regulates the expression of genes associated with metastasis in the AR-negative cell line DU145 compared to the AR-positive cell line LNCaP [306]. Knocking down menin in DU145 cells results in an increase in the expression of metastasis-related genes such as *TIPM2*, *CAPG*, *GSN*, *HTRA1*, and *LAMB3* [306]. However, the effect of menin on the regulation of metastasis-related genes in AR-negative cells may be cell line-dependent, as knocking down menin in AR-negative PC3 cells has no effect on metastasis-related gene expression [306]. Taken together, these results demonstrate that menin suppresses metastasis in AR-negative prostate cancers, at least in part, by regulating a group of genes linked to metastatic pathways in a manner that is context-dependent and not solely dependent on AR status [306].

While menin primarily regulates different genes in AR-positive and AR-negative cells, menin regulates genes associated with anti-growth pathways, like signal transduction and programmed cell death, in both AR-positive and AR-negative cells [306]. Though it is unclear whether the genes are regulated in the same direction, these data suggest that menin regulates some genes in a manner independent of their interaction with AR. Additionally, AR-positive and AR-negative cell lines share at least one binding partner compared to a normal prostate cell line. In both AR-positive and AR-negative cell lines, menin interacts with the transcription factor ERH [305], which has recently been reported to induce cell migration and invasion in urothelial carcinoma through the activation of c-Myc [325]. Whereas in a non-malignant prostate cell line, menin interacts with the transcription factor SNW1, which has been shown to promote TGFβ signaling and negatively regulate growth in the prostate [326,327]. These data suggest that menin interacts with different epigenetic modifiers in prostate cells that have undergone oncogenic transformation and that some of menin’s binding partners are the same regardless of whether AR is expressed.

#### 6.1.4. Menin and HSP27

Menin expression is regulated post-translationally by heat shock protein 27 (HSP27) in AR-negative prostate cancers [305]. HSP27 expression is upregulated in prostate cancer, and HSP27 has previously been established as a promoter of castration-resistant prostate cancer [328,329,330]. Overexpressing HSP27 in AR-positive cells conveys hormone insensitivity and increased menin expression by reducing the ubiquitination of menin and its subsequent proteasomal degradation [305]. Knocking down menin partially recapitulates the effects of HSP27 knockdown on cell proliferation, tumor xenograft growth, and chemotherapeutic sensitivity [305,328,329,330], suggesting that menin mediates HSP27’s pro-tumorigenic effects. Notably, Bourefis et al. found that using a combination of HSP27 and menin serum levels was more accurate for predicting prostate cancer aggressiveness and mortality than PSA [331], suggesting that high serum menin levels may be useful for identifying individuals with prostate cancer who may be at increased risk for aggressive disease and mortality.

#### 6.1.5. Menin in Murine Prostate Cancer

Interestingly, menin seems to function differently in human and murine prostates. Though menin expression is upregulated in human prostate cancers, mouse studies indicate that menin suppresses the initiation of prostate cancer in mice [307,332]. A study by Teinturier and colleagues found that menin knockout mice develop prostatic intraepithelial neoplasia (PIN) more quickly than menin wild-type mice [307]. Furthermore, in the menin knockout mice, some mice progressed to microinvasive adenocarcinomas by 6 months of age, while none of the wild-type mice progressed to microinvasive adenocarcinomas by 10 months of age [307]. The early-stage PIN lesions in the menin knockout mice were more proliferative and had lower expression of AR, multiple AR target genes, the epithelial marker cytokeratin 18, and E-cadherin compared to menin wild-type mice. This suggests that loss of menin results in loss of differentiation in murine prostate luminal cells [307]. Additionally, heterozygous inactivation of *Men1* increased the incidence of PIN, as well as both prostate adenocarcinomas and in situ carcinomas, compared to age-matched menin wild-type littermate controls [332]. The tumors that developed in the *Men1*^+/−^ mice had no detectable menin expression, further supporting the notion that menin inhibits prostate cancer progression in mice [332]. Menin suppresses prostate cancer in mice by helping prostate epithelial cells maintain their differentiated state and by reducing cell cycle progression. The tumors that developed in the heterozygous menin knockout mice maintained some AR expression but had reduced expression of the basal epithelial cell marker p63, indicating a loss of differentiation and normal prostate architecture following the loss of menin. These tumors also had decreased expression of the cell cycle inhibitor p27 [332], which is a known target of menin and is frequently inactivated in other cancers [24,27]. While the mouse data conflict with most human studies, the mouse data are consistent with a study by Teinturier and colleagues, which found that menin expression was generally lower in primary hormone-sensitive prostate cancers compared to benign prostate tissue in their cohort of human prostate cancer patients [307]. It is unclear why menin regulates prostate cancer differently in mice compared to humans, and further research is needed to determine the biological mechanisms underlying these differences.

### 6.2. Ewing Sarcoma

Menin is overexpressed in Ewing sarcoma tumors compared to benign adult tissues from various organs [333]. Menin expression was also found to be higher in Ewing sarcoma cell lines compared to bone marrow-derived mesenchymal cells and non-transformed fibroblasts [333]. Inhibiting menin, either genetically with shRNAs or pharmacologically with MI-503, in Ewing sarcoma cells decreases cell proliferation, cell viability, colony formation, and tumor xenograft growth [333]. Menin knockdown in multiple Ewing sarcoma cell lines also reduces the expression of *HOXD13* and *HOXD10* [333], both of which are frequently overexpressed in Ewing sarcomas [334,335]. Dysregulation of Hox genes promotes tumorigenesis in multiple cancers [336], suggesting that menin may promote Ewing sarcoma pathogenesis through its regulation of *HOXD* gene expression.

Menin may also contribute to Ewing sarcoma tumorigenesis by regulating serine and glycine synthesis. Both genetic and pharmacologic inhibition of menin significantly downregulates the serine synthesis pathway (SSP) in multiple Ewing sarcoma cell lines [337]. The SSP is responsible for the de novo synthesis of serine and glycine, which are important intermediates of metabolites necessary for a multitude of key cellular processes, including nucleotide synthesis, ATP synthesis, phospholipid synthesis, maintenance of redox homeostasis, DNA methylation, and histone methylation [338,339,340,341,342]. Dysregulation of the SSP is a hallmark of multiple cancers [338,342]. In Ewing sarcoma specifically, the oncoprotein EWS-FLI has been shown to upregulate SSP gene expression and consequently promote tumorigenesis [343,344,345]. Furthermore, PHGDH, which catalyzes the rate-limiting step of the SSP, is overexpressed in Ewing sarcoma and promotes Ewing sarcoma cell proliferation, survival, and tumor growth [337].

Notably, the downregulation of the SSP in Ewing sarcoma cells following menin inhibition is not simply a result of decreased cell proliferation or glucose uptake. Alternative methods of slowing proliferation, such as serum starvation and DNA methyltransferase inhibitors, have minimal effect on SSP gene expression, and inhibiting menin with MI-503 has no effect on glucose uptake [337]. Instead, menin inhibition reduces activating H3K4me3 marks at the promoter of *PHGDH* [337], decreases activating transcription factor 4 (ATF4) expression [346], and decreases ATF4 localization at the promoters of SSP genes *PHGDH* and *PSAT1* [346]. ATF4 is a master regulator of transcription for amino acid metabolism and stress responses [338,347,348,349] and has been shown to activate the SSP in various solid tumors that lack amplification of the *PHGDH* locus [347,350,351,352,353], such as Ewing sarcoma [337]. These data, therefore, indicate that menin regulates the SSP by promoting the expression of SSP genes through ATF4 and the menin–MLL complex.

While there is ample evidence that menin promotes the activation of the SSP in Ewing sarcoma cell lines, additional studies are needed in more physiologically relevant models of Ewing sarcoma to better understand how menin contributes to Ewing sarcoma tumorigenesis. Further research is also needed to fully characterize the mechanisms through which menin regulates the SSP in Ewing sarcoma. For example, while EWS-FLI has been shown to regulate ATF4 expression, EWS-FLI expression is not affected by menin inhibition [346], indicating that menin likely regulates ATF4 through mechanisms independent of EWS-FLI. Additionally, while menin inhibition reduces H3K4me3 at SSP gene promoters, knocking down MLL does not have as robust an effect on SSP gene expression as menin knockdown [337], suggesting that menin may regulate SSP promoter methylation through additional mechanisms independent of MLL. Multiple metabolic pathways, in addition to the SSP, were dysregulated upon menin inhibition [337], and future studies are necessary to determine whether any of these pathways play a functional role in Ewing sarcoma pathobiology. Furthermore, additional studies are needed to better understand the efficacy of menin inhibitors in slowing Ewing sarcoma tumorigenesis. While MI-503 has been shown to decrease Ewing sarcoma cell proliferation and tumor growth, VTP50469, a menin inhibitor that displays significant tumor-suppressive activity in leukemia, had only a modest effect on Ewing sarcoma tumor growth in a xenograft mouse model [354], indicating that only specific menin inhibitors may be effective for treating Ewing sarcoma.

### 6.3. Lung Adenocarcinoma

Menin expression is lower in primary lung adenocarcinomas compared to adjacent non-malignant lung tissue, and lymph node metastasis is more common in individuals with lung adenocarcinomas harboring low menin expression compared to those with moderate to high menin expression [70].

Menin expression is downregulated in lung adenocarcinoma, at least partially through KRAS, a major driver of lung tumorigenesis [355], and the microRNA miR-802 [356]. Menin expression is inversely correlated with RAS expression in lung adenocarcinomas [357], and KRAS increases DNA methylation at the *MEN1* promoter by upregulating DNMT1/3B and increasing DNMT1 recruitment to the menin promoter [357]. Meanwhile, miR-802 recognizes a sequence in the 3′-UTR of menin and is overexpressed in lung carcinomas, resulting in the degradation of menin transcripts and a subsequent decrease in menin protein levels in lung adenocarcinoma [356]. miR-802 promotes cell proliferation in lung adenocarcinoma cells, presumably as a result of decreased menin expression, though it is unclear whether the increase in proliferation is dependent on a decrease in menin expression [356]. Overexpressing the miR-802 precursor in lung adenocarcinoma lines alters the expression of known menin target genes. Specifically, miR-802 decreases p18 and p27 expression and increases β-catenin and NF-κB (p65) expression [356], which is consistent with miR-802 reducing menin protein expression [210,239,268].

#### 6.3.1. Suppression of Pleiotrophin

Menin inhibits lung adenocarcinoma proliferation, migration, and tumor growth in mice in part by suppressing the expression of the growth factor pleiotrophin (PTN) [69,70], similar to its effect in pNETs [68]. PTN is frequently overexpressed in lung cancer [358]. Unlike in leukemias and several types of solid tumors, where menin alters gene expression through menin–MLL complex-mediated histone modification, menin has no effect on H3K4 methylation or H3 acetylation at the *PTN* promoter in lung adenocarcinoma [70]. Instead, menin recruits the polycomb repressive complex 2 (PRC2) to the *PTN* promoter and reduces RNA polymerase II binding, thereby increasing the deposition of transcriptionally repressive H3K27me3 marks and decreasing *PTN* transcription [70]. Inhibition of menin increases cell proliferation and PTN expression [70], while overexpression of PTN completely ablates menin’s antiproliferative and antimigratory effects [69,70]. Mechanistically, PTN activates FAK, PI3K, and ERK signaling through the activation of its receptor RPTPβ/ζ [69]. Inhibiting PI3K or MEK1/2 suppresses cell migration in a manner similar to menin overexpression, suggesting that menin suppresses the activation of PI3K, ERK, and FAK signaling downstream of PTN activation [69].

#### 6.3.2. Regulation of p53

Menin also suppresses lung tumorigenesis through the positive regulation of p53. Genetically depleting menin in lung adenocarcinoma cells and lung tissue of mice decreases p53 expression [108]. Mechanistically, menin promotes p53 expression by binding to and increasing H3K4 trimethylation at the promoter of the E3 ligase βTrCP [108], presumably through the recruitment of MLL. This downregulation of βTrCP by menin decreases the ubiquitylation and subsequent degradation of p53 in lung adenocarcinoma cells [108]. The decrease in p53 expression upon loss of menin results in increased cell cycle progression, as well as DNA damage, as measured by increased γH2AX positivity [108]. Interestingly, while loss of menin increases γH2AX positivity in lung adenocarcinoma cells, consistent with downregulation of p53, it also increases activating phosphorylation of ATM, and the expression of the homologous recombination (HR) protein RAD51 and the non-homologous end joining (NHEJ) proteins Ku70/Ku80 [108]. These data suggest that while menin protects against DNA damage in the lung through positive regulation of *TP53* gene expression, loss of menin triggers both the p53-dependent HR and p53-independent error-prone NHEJ DNA repair mechanisms, perhaps in response to the increase in double-strand breaks.

#### 6.3.3. Regulation of Alternative Splicing

Menin further suppresses lung tumorigenesis through the regulation of alternative splicing [359]. In mouse lung tissue and lung adenocarcinoma cell lines, genetically depleting menin results in aberrant splicing of genes involved in RNA metabolism and mRNA splicing [359]. A subset of these aberrant splicing events correlates with patient survival in lung adenocarcinoma, while the expression of the genes themselves is not associated with survival [359]. These data indicate that menin is a key regulator of alternative splicing in lung adenocarcinoma and that dysregulation of splicing can contribute to lung adenocarcinoma pathobiology. While menin does not directly interact with members of the spliceosome, menin slows RNA polymerase II elongation, which allows the splicing machinery to recognize variant exons with weaker splice sites [359,360]. Knocking out menin increases the rate of transcription by RNA polymerase II and inhibits the interaction between RNA polymerase II and the splicing factors SRSF2 and U2AF65 [359]. This reduced recruitment of splicing factors to the nascent RNA results in prolonged interaction between the naked RNA and the single-stranded DNA template, a situation prone to R-loop formation [359,361]. Unintended R-loop formation has been shown to cause DNA damage and genome instability, which can contribute to tumorigenesis [362,363,364]. In lung adenocarcinoma specifically, knocking out menin in lung adenocarcinoma cells increases R-loop formation and significantly increases the number of γH2AX positive nuclei, indicating increased DNA damage. Treating menin knockout cells with an RNase that specifically degrades RNA–DNA hybrids, like R-loops, blocks the increase in γH2AX positivity, indicating that the increased DNA damage caused by loss of menin is a result of increased R-loop formation [359]. Notably, genetically depleting menin sensitizes lung adenocarcinoma cells to splicing inhibitors [359]. Combination treatment with menin inhibitors and splicing inhibitors may, therefore, merit investigation as a potential lung adenocarcinoma therapy, and splicing inhibitors alone might be a viable treatment strategy for individuals with lung adenocarcinomas expressing low levels of menin. Importantly, lung adenocarcinoma is the only tumor type in which menin has been found to regulate alternative splicing, potentially indicating a unique role for menin in the lung.

#### 6.3.4. Regulation of Immunosurveillance

Menin can also inhibit lung tumorigenesis by promoting the immunogenicity of lung adenocarcinomas [365]. Genetic depletion of menin in a *Kras^G12D^*-driven lung adenocarcinoma mouse model accelerates tumor formation. This increase in tumor growth following the loss of menin is accompanied by significant dysregulation of multiple immune-related pathways and a decreased percentage of tumor-infiltrating cytotoxic T-cells [365]. Mechanistically, menin promotes immune cell infiltration into tumors by downregulating the expression of programmed cell death-ligand 1 (PD-L1). Genetic and pharmacologic inhibition of menin in lung adenocarcinoma cell lines and the *Kras^G12D^*-driven tumor model results in significant upregulation of PD-L1 while overexpressing menin in lung adenocarcinoma cell lines inhibits PD-L1 expression [365]. PD-L1 expression is upregulated in many cancers, and overexpression of PD-L1 is a key mediator of T-cell exhaustion and tumor immune escape (reviewed in [366]). Menin downregulates PD-L1 expression by suppressing the expression of the deubiquitinating enzyme CSN5, which has previously been shown to target PD-L1 [367,368], and by inhibiting the interaction between PD-L1 and CSN5, thereby promoting the proteasomal degradation of PD-L1 [365]. In a cohort of individuals with lung adenocarcinoma, menin expression was negatively correlated with PD-L1 expression but positively correlated with CD3 and CD4 positivity [365]. In addition, a high ratio of tumor menin to PD-L1 expression was associated with improved survival [365], further suggesting that menin functions as a key mediator of immune infiltration in lung adenocarcinoma.

#### 6.3.5. Suppression of RAS Signaling

Menin can further inhibit lung adenocarcinoma tumorigenesis through the suppression of RAS signaling. Menin reduces the binding of the guanine exchange factors (GEFs) GRB2 and SOS1 to RAS in lung cancer [357]. Interestingly, knocking out menin in the lung accelerates Kras^G12D^-induced tumor formation in the *Men1^f/f^;Kras^G12D/+^;Ubc9-Cre-ER* mouse model [357]. Additionally, deletion of menin in the lungs of a different Kras^G12D^-driven mouse model of lung tumorigenesis (*LSL*-*Kras*^G12D/+^;*Sftpc*-Cre) increases tumor expression of Hmga2, a downstream target of Kras signaling [369] and a marker of invasion and metastasis [108]. However, a study by Zhu and colleagues found that menin may also function as a tumor promoter in KRAS-driven lung cancer in some contexts. Treating lung adenocarcinoma cells and tumor xenografts with MI-3 inhibits the growth of KRAS-mutant cell lines and tumors with minimal effect on the growth of wild-type KRAS lines and tumors [370]. Genetically depleting menin in the lungs of mice similarly suppresses tumor growth in a Kras^G12D^ model of lung tumorigenesis [370]. Furthermore, loss of menin in an *LSL*-*Kras*^G12D/+^;*Sftpc*-Cre mouse model of lung cancer increases the expression of the epithelial marker E-cadherin and reduces the expression of the mesenchymal markers Nestin, Vimentin, and ZEB1 [108], suggesting that menin promotes epithelial-to-mesenchymal transition (EMT) and metastasis of lung tumors. These conflicting data indicate that additional research is needed to define the contexts in which menin may switch from a tumor suppressor to a tumor promoter in lung adenocarcinoma. Additionally, as there is currently only one study demonstrating that menin can promote lung tumorigenesis, further investigation is warranted to better characterize menin’s function as a tumor promoter in lung adenocarcinoma.

#### 6.3.6. Activation of Kras-Mediated Oncogene-Induced Senescence

Menin may also suppress lung tumorigenesis by triggering senescence in oncogenically transformed lung tissue. Activation of the Kras-mediated oncogene-induced senescence program is a key barrier to the initiation of lung cancer [371]. Loss of menin in the lungs of *LSL*-*Kras*^G12D/+^;*Sftpc*-Cre mice significantly reduces senescence, as measured by decreased SA β-gal staining, decreased expression of the critical oncogene-induced senescence mediator p16, and decreased senescence-associated H3K9me3 heterochromatic foci [108]. Senescence-associated secretory profile (SASP) genes, including TGFβ and IL-6, are also significantly reduced following the loss of menin, and inhibiting menin with MI-3 in an induced-senescence cell culture model similarly reduces p16 and SASP gene expression [108]. These data suggest that the loss of menin allows for the initiation of lung tumorigenesis by inhibiting oncogene-induced senescence.

### 6.4. Melanoma

Clinical data suggest that individuals with MEN1 syndrome may be at an increased risk of developing skin tumors (reviewed in [372]), including melanomas [373,374,375,376]. While there are conflicting data regarding whether menin is mutated in melanoma [377,378], mechanistic studies show that menin suppresses melanomagenesis [71,379].

Menin expression is downregulated in malignant melanomas compared to benign nevi [71,379], as well as in melanoma cell lines and short-term melanoma cell cultures compared to primary melanocytes [379]. This downregulation results from increased methylation of CpG islands within the *MEN1* gene promoter and an atypical CpG island located in the intronic region between exons 1 and 2 [71,379]. Consistent with this, Gao et al. found that inhibiting DNA methylation with 5-AZA-dC decreased methylation of the *MEN1* promoter and increased menin expression [71]. Notably, 5-AZA-dC also decreased melanoma cell line proliferation and migration [71], suggesting that menin may suppress melanoma growth and progression.

Overexpressing menin in melanoma cell lines suppresses melanoma growth and invasion both in vitro and in vivo [71,379], further supporting menin’s tumor-suppressive role in melanoma. Similar to its effect in lung adenocarcinoma, menin inhibits melanomagenesis by inhibiting the expression of the growth factor PTN and its cell surface receptor RPTPβ/ζ [71]. PTN is overexpressed in melanoma [380,381], and PTN exerts multiple pro-tumorigenic functions through its activation of RPTPβ/ζ [69,70,382,383,384]. Accordingly, knocking down PTN in melanoma cell lines recapitulates the effect of menin overexpression on tumorigenesis [71]. In addition to inhibiting PTN activity, menin also suppresses melanoma growth and progression through the activation of the TGFβ signaling pathway [385]. The TGFβ signaling pathway acts as a potent suppressor of melanoma [386,387,388,389], and genetic depletion of menin in melanoma cell lines dampens TGFβ’s inhibition of cell cycle progression and induction of apoptosis, as measured by caspase-3 expression [385]. Furthermore, loss of menin partially ablates the downregulation of c-Myc and the increased transcription of TGFβ pathway target genes in melanoma cell lines following stimulation with exogenous TGFβ [385]. These data suggest that menin functions as a downstream effector of TGFβ in melanoma, suppressing melanomagenesis by facilitating cell cycle arrest and apoptosis in response to TGFβ [385]. Furthermore, menin can suppress melanomagenesis by inhibiting PI3K and MAPK signaling, also similar to its effects in pNETs, gastric cancer, ER-positive breast cancers, and lung adenocarcinoma [71]. The ERK/MAPK and PI3K/Akt signaling pathways are key drivers of melanoma growth and progression (reviewed in [390]), and overexpressing menin decreases the protein levels of pFAK, pERK1/2, and PI3K, partially through inhibition of PTN and RPTPβ/ζ activity [71]. Unlike in other cancers where menin regulates β-catenin [243,247,268,310], menin has no effect on β-catenin expression or activation in melanoma cells, suggesting that menin’s tumor-suppressive activity in melanoma is independent of Wnt signaling.

Menin also suppresses melanomagenesis by activating the DNA damage response and promoting DNA double-strand break repair. Melanocytes and melanoma cells upregulate menin in response to DNA damage [71,379] through phosphorylation and subsequent stabilization of the menin protein by the DNA damage kinase ATM [379]. Gao and colleagues found that this upregulation of menin increased apoptosis in melanoma cells treated with cisplatin [71]. Furthermore, overexpressing menin in melanoma cell lines with low baseline menin expression significantly improves their ability to repair DNA double-strand breaks through homologous recombination, while loss of menin in primary melanocytes hampers this ability [379]. Mechanistically, menin facilitates DNA double-strand break repair by promoting transcription of key homologous recombination (HR) genes, including *BRCA1*, *RAD51*, and *RAD51AP1* [379]. Menin is recruited to the promoters of *BRCA1*, *RAD51*, and *RAD51AP1* by ERα, where it then recruits MLL and increases H3K4 trimethylation and expression of BRCA1, RAD51, and RAD51AP1 [379]. Consistent with this, loss of menin in melanocytes decreases the frequency of HR and increases the frequency of error-prone non-homologous end joining (NHEJ) [379]. This shift from homologous recombination to NHEJ suggests that the loss of menin may further promote melanoma tumorigenesis by increasing genomic instability.

Notably, three pathogenic germline variants (PGVs) in *MEN1* that are frequently observed in MEN1 syndrome abolish the upregulation of BRCA1, RAD51, and RAD51AP1 and consequently abolish menin’s ability to promote DNA double-strand break repair [379]. These data suggest that some individuals with MEN1 syndrome may be at increased risk of developing melanoma, and further research is needed to explore this potential risk. Additionally, as menin is recruited to HR target genes by ERα and melanoma incidence and mortality are significantly higher in males, future studies should aim to determine whether menin contributes to the male sex bias in melanoma and whether menin is regulated by estrogen signaling in other cancers where differential sex-based outcomes exist. Further research is also needed to determine the mechanism behind increased *MEN1* methylation. Methylation of *MEN1* is regulated by KRAS in lung adenocarcinoma [357], so it is possible that NRAS, the second most commonly mutated driver gene in melanoma, may similarly be responsible for the increased *MEN1* promoter methylation observed in melanoma. Furthermore, since menin’s function in melanoma shares similarities to its function in lung adenocarcinoma, it would be of interest to explore whether menin also regulates alternative splicing in melanoma and whether the loss of menin may not only impair the DNA damage response but may also increase DNA damage through aberrant R-loop formation similar to lung adenocarcinoma.

### 6.5. Glioma

Menin is overexpressed at the protein level in some adult gliomas [391], with high menin-expressing gliomas being associated with shorter median and overall survival [391]. Tumors with high menin expression also trended to have greater Ki67 positivity compared to low menin-expressing tumors [391], suggesting menin promotes the growth and progression of adult gliomas.

In support of menin functioning as a tumor promoter in gliomas, menin inhibition with MI-3 reduces colony formation and viability in three out of four glioma cell lines tested [391]. MI-3 also decreases Ki67 positivity in the responsive glioma cells [391], suggesting that menin promotes glioma cell proliferation and survival. Using a different inhibitor, MI-2, results in a similar reduction in the proliferation and viability of transformed neural precursor cells (NPCs) that are used as a model for diffuse intrinsic pontine glioma (DIPG) [392], an aggressive form of pediatric glioma [393,394,395,396]. Interestingly, the effects of both MI-3 and MI-2 are specific to glioma cells, with MI-3 having no effect on neuroblastoma cells [391] and MI-2 having no effect on non-transformed NPCs or a less aggressive DIPG model lacking a H3.3K27 mutation [392].

However, subsequent work found that MI-2 may primarily suppress DIPGs and adult gliomas independent of menin [397]. Glioma cells in which menin is genetically depleted using CRISPR/Cas9 are still highly responsive to MI-2, while the structurally distinct MI-503 has no effect on glioma cell viability [397]. MI-2 treatment led to increased cholesterol export and decreased cholesterol biosynthesis, leading to the depletion of intracellular cholesterol that ultimately reduces the viability of glioma cells in a manner independent of menin [397]. Although there is controversy regarding the mechanism of action of MIs in gliomas, these data suggest that MIs may be a viable therapeutic strategy for glioma, including aggressive DIPGs.

Menin depletion with shRNA also increases astrocytic differentiation of the transformed NPCs while having no effect on non-transformed NPCs or the less aggressive DIPG model [392]. Menin expression is normally downregulated during astrocyte differentiation [392]; however, in the transformed NPCs, menin expression is six times higher than in non-transformed NPCs [392], indicating that aberrant menin expression may promote glioma progression by keeping cells in an undifferentiated, stem-like state, in addition to promoting cell proliferation.

There are multiple other mechanisms by which menin may also promote glioma tumorigenesis, including the regulation of Wnt/β-catenin signaling and intracellular interactions. Menin is recruited to Wnt target genes by PYGO2, which is overexpressed in gliomas and is associated with poor survival [398]. In complex with MLL, menin promotes H3K4 trimethylation at the promoters of Wnt target genes, subsequently increasing Wnt target gene transcription [398]. These data suggest that menin may promote glioma tumorigenesis through the activation of Wnt signaling. Menin is also highly expressed in the cytoplasm of glioma cells [399], where it interacts with the type III intermediate filament proteins glial fibrillary acidic protein (GFAP) and vimentin [400], suggesting that menin may also promote migration and metastasis of glioma cells. Future studies are needed to confirm menin’s interaction with GFAP and vimentin in additional glioma cell lines and patient samples, as well as to investigate the effects of menin on glioma cell migration and invasion.

## 7. Promises and Challenges of Menin Inhibitors in Cancer

Menin inhibitors have shown promising therapeutic potential in preclinical models of multiple cancer types (Table 4) and are currently being investigated in clinical trials for acute leukemias and some solid tumor types (Table 1, Table 2 and Table 3). Preliminary results from these early-phase trials show promising safety and efficacy profiles associated with the use of menin inhibitors in individuals with MLLr and NPM1-mutant leukemias (reviewed in [172,174,175]). More specifically, these clinical trials show rapid and durable responses to SNDX-5613 (revumenib) and KO-539 (ziftomenib), with a relatively low occurrence of high-grade adverse events in heavily pretreated individuals with acute leukemias who have failed prior lines of therapy [187,401,402,403,404,405]. Importantly, these promising responses are observed across multiple types of acute leukemias driven by multiple different MLL fusion proteins or NPM1 mutants [401,405].

However, despite the favorable safety and efficacy profiles, there are some potential drawbacks to menin inhibitors. For example, severe adverse events have occurred in a small proportion of individuals receiving SNDX-5613 and KO-539 [187,401,402,403,404,405], with a small percentage of these adverse events not being able to be reversed through dose reduction [401]. In addition, some individuals acquire resistance to menin inhibitors, limiting the duration of their response [186,187], and it remains unclear why some individuals respond better than others. Furthermore, there are concerns that inhibiting menin may promote unintended growth and/or cancer development in tissues where menin typically functions as a tumor suppressor. Since menin inhibitors are currently only in the early phases of clinical trials, extended follow-ups on individuals receiving menin inhibitors are necessary to determine whether these inhibitors have any meaningful effects on tumorigenesis in these tissues where menin serves a tumor suppressive role.

## 8. Conclusions

In the nearly three decades since menin was first discovered and identified as a potent suppressor of neuroendocrine tumorigenesis, our knowledge of the functions of menin has expanded greatly, with menin now appreciated to play a role in the development of a myriad of solid and hematologic cancers. Notably, the role played by menin in tumorigenesis varies depending on the tumor type and, in some tumors, the specific context. While menin has been recognized as a suppressor of neuroendocrine tumor growth for decades, menin is increasingly appreciated to also suppress tumorigenesis in the bile ducts, exocrine pancreas, stomach, lungs, and melanocytes of the skin. On the other hand, menin is also increasingly recognized to promote tumorigenesis in the blood, colon, ovaries, endometrium, bone/soft tissue, and glial cells of the central nervous system. Moreover, more recent data demonstrate that menin can either suppress or promote breast and prostate tumorigenesis depending on hormone receptor status and may also have dual functions in the liver. Nevertheless, it remains unclear precisely how and why menin suppresses some types of solid tumors yet promotes the development of others. Given menin’s many emerging and important roles across different cancers, as well as the increasing amount of menin-focused cancer research and cancer-focused clinical trials of MIs, the next decade of menin research in cancer will undoubtedly be enlightening.

## Figures and Tables

**Figure 1 genes-15-01231-f001:**
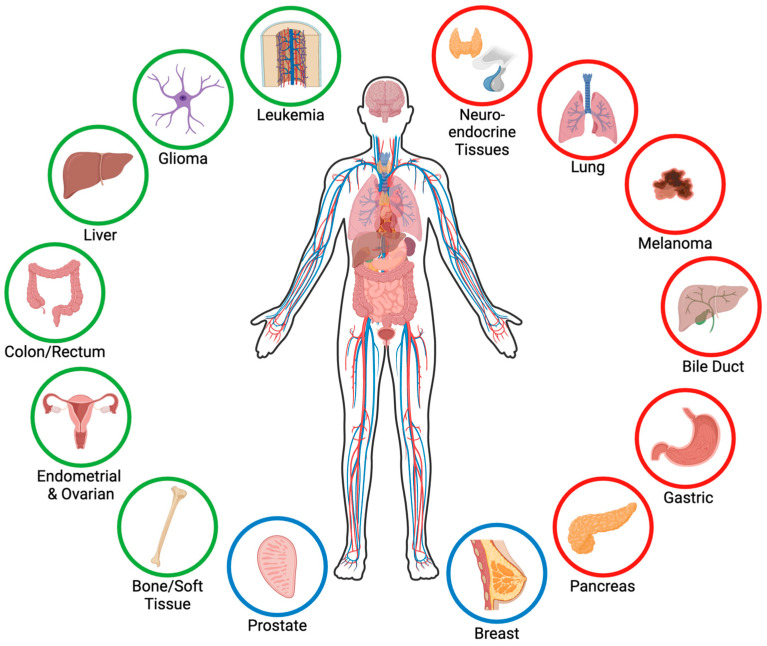
The tissue-specific roles of menin in cancer. In the nearly three decades since menin was first discovered and identified as a potent suppressor of neuroendocrine tumorigenesis, the field of menin in cancer has expanded greatly. Menin is now understood to play a role in the development of a broad range of solid and hematologic cancers. Notably, the role played by menin in tumorigenesis varies depending on the tumor type and, in some tumors, the specific context. While menin has been recognized as a suppressor of neuroendocrine tumor growth for decades, menin is increasingly appreciated for its function as a tumor suppressor (red circle) in the bile ducts, exocrine pancreas, stomach, lungs, and melanocytes of the skin. On the other hand, menin is also increasingly recognized to function as a tumor promoter (green circle) in the blood, colon, ovaries, endometrium, bone/soft tissue, and glial cells of the central nervous system. Menin can also either suppress or promote breast and prostate tumorigenesis depending on hormone receptor status and may have dual functions in the liver (blue circle). Though it remains unclear precisely how menin suppresses some types of solid tumors while promoting the development of others, the striking increase in the amount of menin-focused cancer research over the past decade promises advancements that will continue to expand our understanding of menin biology. The next decade of menin research in cancer is sure to be enlightening. Created with BioRender.com.

**Table 1 genes-15-01231-t001:** Single agent small molecule menin inhibitor clinical trials for leukemia from ClinicalTrials.gov.

Trial	Drug	Phase	Start Date	Sponsor	Cancer Patient Population	Status/ Accrual/ Sites
NCT-04067336	Zifto-menib	I/II	12 September 2019	Kura Oncology	Adults 18+ R/R MLLr (KMT2Ar) and NPM1 mutant AML	Actively- recruiting 199 individuals 56 sites
NCT-04065399	SNDX-5613 (Revu-menib)	I/II	5 November 2019	Syndax Pharma-ceuticals	Adults, children, and infants 30 days+ R/R MLLr and NPM1 mutant AML/ALL/ALAL/MLL/MPL	Actively- recruiting 413 individuals 52 sites
NCT-04811560	JNJ-75276617	I/II	19 May 2021	Janssen	Adults and children 12+ R/R MLLr and NPM1 mutant AML and ALL	Actively- recruiting 150 individuals 28 sites
NCT-05153330	BMF-219	I	24 January 2022	Biomea Fusion	Adults 18+ MLLr and NPM1 mutant AML/ALL/DLBCL/MM/CLL/SLL	Actively- recruiting 177 individuals 45 sites
NCT-04988555	DSP-5336	I/II	28 February 2022	Sumitomo Pharma America	Adults 18+ R/R AML and ALL with and without MLLr and NPM1 mutation	Actively- recruiting 70 individuals 15 sites
NCT-06052813	BN104	I/II	16 September 2023	BioNova Pharma-ceuticals (Shanghai)	Adults 18+ R/R AML and ALL	Actively- recruiting 90 individuals 1 site
NCT-06229912	SNDX-5613	II	31 July 2024 (Est.)	M.D. Ander-son Cancer Center	Adults and children 12+ R/R AML and ALL with genetic alterations associated with upregulation of Hox genes	Actively- recruiting 15 individuals 1 site

Abbreviations: ALAL, acute leukemia of ambiguous lineage; ALL, acute lymphocytic leukemia; AML, acute myeloid leukemia; AUL, acute undifferentiated leukemia; Est., estimated; MLLr, mixed lineage leukemia rearranged; MPL, myeloproliferative leukemia; NPM1, nucleophosmin 1; R/R, relapsed/refractory.

**Table 2 genes-15-01231-t002:** Menin inhibitor combination therapies in clinical trials for leukemia from ClinicalTrials.gov.

Trial	Drug	Phase	Start Date	Sponsor	Cancer Patient Population	Status/ Accrual/ Sites
NCT-04752163	DS-1594b with or without azacitidine, venetoclax, or mini-HCVD	I/II	25 March 2021	M.D. Anderson Cancer Center	Adults 18+ R/R AML and ALL	Completed 8 November 2023 17 individuals 1 site
NCT-05326516	SNDX-5613 with chemo	I	9 March 2022	Syndax Pharma-ceuticals	Infants to adults (30 days+) R/R AML/ALL/ALAL/AUL	No longer recruiting 30 individuals 15 sites
NCT-05360160	SNDX-5613 with decitabine/cedazur-idine (ASTX727) and venetoclax	I/II	14 October 2022	M.D. Anderson Cancer Center	Adults and children 12+ AML	Actively- recruiting 43 individuals 1 site
NCT-05453903	JNJ-75276617 with AML directed therapies	I	4 October 2022	Janssen	Adults 18+ R/R or newly diagnosed MLLr and NPM1 mutant AML	Actively- recruiting 150 individuals 34 sites
NCT-05761171	SNDX-5613 with chemo	II	8 January 2024	Children’s Oncology Group	Children and infants (1 mo–6 yrs) R/R MLLr ALL/ALAL/MPAL	Actively- recruiting 78 individuals 36 sites
NCT-06222580	SNDX-5613 with gilteritinib	I	20 February 2024	Ohio State University Compre-hensive Cancer Center	Adults 18+ R/R FLT3-mutated AML with MLLr or NPM1 mutation	Actively- recruiting 30 individuals 1 site
NCT-05886049	SNDX-5613 with daunorubicin and cytarabine	I	20 June 2024	National Cancer Institute	Adults 18–75 MLLr and NPM1 mutant treatment naïve AML	Actively- recruiting 28 individuals 4 sites
NCT-06376162	Zifto-menib with fludarabine and cytarabine	I	June 2024 (Est.)	Leukemia and Lympho-ma Society	Infants to young adults (0–21) R/R MLLr, NPM1 mutant, and NUP98r pediatric AML and ALL	Not yet recruiting Est. 20 individuals TBD sites
NCT-06284486	SNDX-5613 and venetoclax	II	31 July 2024 (Est.)	M.D. Anderson Cancer Center	Adults and children 12+ MRD AML	Not yet recruiting Est. 8 individuals 1 site
NCT-06313437	SNDX-5613 with 7+3 chemo and mido-staurin	I	September 2024 (Est.)	Dana-Farber Cancer Institute	Adults 18–75 AML newly diagnosed and treatment naïve (excluding hydroxyurea treatment)	Not yet recruiting 22 individuals 2 sites
NCT-06448013	Zifto-menib with venetoclax and gemtu-zumab	I	29 November 2024 (Est.)	M.D. Anderson Cancer Center	Children and young adults (2–21) Pediatric AML	Not yet recruiting Est. 22 individuals 1 site
NCT-05521087	JNJ-75276617 with chemo	I	26 December 2025 (Est.)	Janssen	Infants to adults (30 days–30 yrs) R/R MLLr, NPM1 mutant, and NUP98r acute leukemias	Not yet recruiting 80 individuals 15 sites

Abbreviations: ALAL, acute leukemia of ambiguous lineage; ALL, acute lymphocytic leukemia; AML, acute myeloid leukemia; AUL, acute undifferentiated leukemia; Est., estimated; MLLr, mixed lineage leukemia rear-ranged; NPM1, nucleophosmin 1; R/R, relapsed/refractory.

**Table 3 genes-15-01231-t003:** Single agent small molecule menin inhibitor clinical trials for solid tumors from ClinicalTrials.gov.

Trial	Drug	Phase	Start Date	Sponsor	Cancer Patient Population	Status/ Accrual/ Sites
NCT-05631574	BMF-219	I	12 January 2023	Biomea Fusion	Adults 18+ KRAS driven NSCLC, PDAC, and CRC	Actively- recruiting 90 individuals 27 sites
NCT-05731947	SNDX-5613	I/II	4 April 2023	Syndax Pharma-ceuticals	Adults 18+ CRC and other solid tumors that have failed at least one prior line of therapy	Actively- recruiting 158 individuals 10 sites

Abbreviations: NSCLC, non-small cell lung cancer; PDAC, pancreatic ductal adenocarcinoma; CRC, colorectal cancer.

**Table 4 genes-15-01231-t004:** Menin inhibitors studied in cancer.

Cancer	Menin Inhibitor	
Gastroenteropancreatic NETs	MI-503	
Leukemia	MI-2	VTP50469
	MI-2-2	SNDX-5613 *
	MI-3	KO-539 *
	MI-463	JNJ-75276617 *
	MI-503	DS-1594b *
	MI-538	BMF-219 *
	MI-3454	DSP-5336 *
	MI-1481	BN104 *
Hepatocellular Carcinoma	MI-1	
	MI-3	
	MI-503	
Pancreatic Ductal Adenocarcinoma	BMF-219 *	
Colorectal Cancer	MI-2-2	
	MI-463	
	MI-503	
	SNDX-5613 *	
	BMF-219 *	
Breast Cancer	MI-2	
	MI-2-2	
	MI-136	
	MI-503	
	MI-3454	
Ovarian Cancer	MI-136	
	MI-503	
Endometrial Cancer	MI-136	
	MI-463	
Prostate Cancer	MI-136	
	MI-503	
Ewing Sarcoma	MI-503	
	MI-3454	
	VTP50469	
Lung Adenocarcinoma	MI-3	
Glioma	MI-2	
	MI-3	
	MI-503	

* denotes an inhibitor used in clinical trials; Abbreviations: NET, neuroendocrine tumor.

## Data Availability

No new data were created or analyzed in this study. Data sharing is not applicable to this article.

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
