# Peer review of "Menin in Cancer"

_genes, 2024, doi:10.3390/genes15091231_

Round 1

Reviewer 1 Report

Comments and Suggestions for Authors

Majer and coauthors submitted the manuscript “Menin in cancer.” The manuscript is a review, where the authors focused on the role of Menin in cancer. Menin has no catalytic activity, but it plays a crucial role as a nuclear scaffold protein, regulating gene expression through its interaction with and regulation of chromatin modifiers and transcription factors. 

Despite the short title, the authors have done an excellent work, describing the knowledge currently available on Menin in different tumors. I have nothing else to add other than a small observation:

The action of Menin is also correlated with that of telomerase and the maintenance of telomeres. Authors can add some information regarding this aspect.

Author Response

Comments 1: Majer and coauthors submitted the manuscript “Menin in cancer.” The manuscript is a review, where the authors focused on the role of Menin in cancer. Menin has no catalytic activity, but it plays a crucial role as a nuclear scaffold protein, regulating gene expression through its interaction with and regulation of chromatin modifiers and transcription factors. 

Despite the short title, the authors have done an excellent work, describing the knowledge currently available on Menin in different tumors. I have nothing else to add other than a small observation:

The action of Menin is also correlated with that of telomerase and the maintenance of telomeres. Authors can add some information regarding this aspect.

Response 1: Thank you for the kind words about our manuscript and for the helpful comments on how to improve it. We have added a paragraph to the colorectal cancer section of the manuscript (section 4.5, page 20-21 lines 824-846) discussing menin’s role in telomere maintenance, focusing on menin’s regulation of TERT transcription in the colon, as that information had the greatest relevance to our review of menin in cancer.

Reviewer 2 Report

Comments and Suggestions for Authors

The manuscript provides a comprehensive review of the role of menin in cancer, highlighting its dual function as both a tumor suppressor and promoter depending on the tumor type and context. The scope is extensive, covering various types of cancers including neuroendocrine tumors, leukemia, and hepatocellular carcinoma, among others. The review is well-structured and informative, making a significant contribution to the understanding of menin's complex roles in cancer biology.

However, below points may be considered for the improvements.

1. While the manuscript covers many cancer types, some sections, such as the role of menin in parathyroid and lung NETs, could benefit from further elaboration. Including more recent studies and discussing conflicting data in these areas would strengthen these sections.

2. Although the review discusses the role of menin in various signaling pathways, the mechanistic details could be expanded in some areas. For instance, the interaction between menin and chromatin modifiers could be explored in greater detail, especially in the context of leukemia.

3. The section on therapeutic potential could include more discussion on the challenges of targeting menin in a clinical setting. Additionally, exploring the potential side effects and the specificity of menin inhibitors could provide a more balanced view.

Author Response

The manuscript provides a comprehensive review of the role of menin in cancer, highlighting its dual function as both a tumor suppressor and promoter depending on the tumor type and context. The scope is extensive, covering various types of cancers including neuroendocrine tumors, leukemia, and hepatocellular carcinoma, among others. The review is well-structured and informative, making a significant contribution to the understanding of menin's complex roles in cancer biology.

However, below points may be considered for the improvements.

Comments 1: While the manuscript covers many cancer types, some sections, such as the role of menin in parathyroid and lung NETs, could benefit from further elaboration. Including more recent studies and discussing conflicting data in these areas would strengthen these sections.

Response 1: Thank you for your insightful comments and suggestions. We have performed additional literature searches focusing on parathyroid tumors and lung NETs, but were only able to identify one additional paper that we did not previously include in the review. We have added a couple of sentences mentioning the findings of this additional paper in the lung NET section (page 8, lines 388-396), but given overall concerns about the length of the manuscript, we are hesitant to expand further on the descriptions of the papers cited in these sections. Nevertheless, if you feel there are specific studies we missed in our literature search, we and would be happy to include any additional studies you send our way.

Comments 2: Although the review discusses the role of menin in various signaling pathways, the mechanistic details could be expanded in some areas. For instance, the interaction between menin and chromatin modifiers could be explored in greater detail, especially in the context of leukemia.

Response 2: Thank you for your comment. Given the concerns about the overall length of the manuscript, we again are hesitant to expand further on the mechanistic details. There are multiple comprehensive reviews that have been published that do a nice job discussing the interaction between menin and chromatin modifiers in the context of leukemia in great detail, and we have further highlighted these reviews in the text (page 10, lines 473-474 and lines 485-486).

Comments 3: The section on therapeutic potential could include more discussion on the challenges of targeting menin in a clinical setting. Additionally, exploring the potential side effects and the specificity of menin inhibitors could provide a more balanced view.

Response 3: Thank you for this insightful suggestion. We have added a paragraph to the end of the manuscript (section 7, pages 34-35, lines 1525-1550) discussing the potential pitfalls of menin inhibitors.

Reviewer 3 Report

Comments and Suggestions for Authors

The authors review the role of menin in both hematological cancers and solid tumors, as well as the feasibility of menin as a druggable target against the diseases. Overall, I find the review is well written and provide comprehensive information.

I suggest the authors revise the structure of the manuscript. The authors should first elaborate the roles of menin in various cancers. Subsequently, the authors discuss menin as a potential druggable target for cancer treatment. This minor amendment is required to improve the readability of the manuscript.

Comments on the Quality of English Language

I would suggest minor editing of English is required. 

Author Response

Comments 1: The authors review the role of menin in both hematological cancers and solid tumors, as well as the feasibility of menin as a druggable target against the diseases. Overall, I find the review is well written and provide comprehensive information.

I suggest the authors revise the structure of the manuscript. The authors should first elaborate the roles of menin in various cancers. Subsequently, the authors discuss menin as a potential druggable target for cancer treatment. This minor amendment is required to improve the readability of the manuscript.

Response 1: Thank you for your thoughtful review of our manuscript. We have carefully considered your suggestion to revise the structure of the manuscript. Our goal is for this manuscript to be a resource for researchers investigating the role of menin in specific cancers. To achieve this goal, we feel it is more appropriate to maintain the current structure of the manuscript to allow for readers to more easily learn about all aspects of their specific cancer of interest. However, we do agree that more discussion of menin as a druggable target is warranted, and therefore we added a section to the end of the manuscript (section 7, pages 34-35, lines 1525-1550) discussing the both the benefits and potential pitfalls of using menin inhibitors to treat cancer.

Reviewer 4 Report

Comments and Suggestions for Authors

This review is important for the field of menin in cancer research. Generally the review article is written well.

Comments:

1. Please create a Table to summarize what menin inhibitors for what cancers.

2. Any information of menin inhibitor for MEN1/2?

3. Any information of menin pathway in MDM2?

4. The Table is not well organized. 

5. In Table 2: Menin inhibi- is truncated.

6. Please list all of the abbreviations.

Author Response

This review is important for the field of menin in cancer research. Generally the review article is written well.

Comments:

Comment 1: Please create a Table to summarize what menin inhibitors for what cancers.

Response 1: Thank you for your kind review of our manuscript and for your thoughtful comments. We have created a new table (Table 4, page 36) that summarizes which menin inhibitors have been studied in the various cancers.

Comments 2: Any information of menin inhibitor for MEN1/2?

Response 2: We appreciate this comment. To our knowledge there is one study in a preclinical mouse model of gastroenteropancreatic NETs which found that the small molecule menin inhibitor MI-503 decreased hypergastrinemia and gastric hyperplasia by stabilizing mutant menin proteins. We mentioned this study’s findings in the Luminal Gastrointestinal NETs section of our manuscript, and we have now expanded on this to discuss the potential for using small molecule menin inhibitors as a therapeutic strategy for MEN1 syndrome (page 9, lines 439-445). We did not discuss MEN2 syndrome, as MEN2 syndrome is the result of germline mutations in RET, not MEN1, and therefore would not be expected to be impacted by menin inhibitors.

Comments 3: Any information of menin pathway in MDM2?

Response 3: We appreciate this inquiry, however after conducting a literature search, we were unable to find any studies identifying any interplay between menin and MDM2 in cancer.

Comments 4: The Table is not well organized. 

Response 4: Thank you for pointing this out. To improve their readability, we rearranged the order of the columns in the tables and ensured that there is no longer any overlap between the different tables.

Comments 5:  In Table 2: Menin inhibi- is truncated.

Response 5: Thank you for pointing this out. We also reformatted the tables so that there is no longer any overlap between the different tables leading to text truncation.

Comments 6: Please list all of the abbreviations.

Response 6: We appreciate this suggestion. We have defined all abbreviations in each table, and we also went back through the manuscript and made sure that all abbreviations are defined in the text the first time they are used. However, as Genes manuscript formatting does not include an abbreviations section, we did not include a list of all abbreviations in the text.

Round 2

Reviewer 4 Report

Comments and Suggestions for Authors

The authors answered my questions. No more comments.